# Mercury Methylation in paddy soil: Source and distribution of mercury species at a Hg mining area, Guizhou Province, China

L. Zhao[1,2], C.W.N. Anderson[3], G.L. Qiu[2], B. Meng[2], D.Y. Wang[1], and X. B. Feng[2]

[1]College of Resources Environment, Southwest University, Chongqing 400716, P. R. China

[2]State Key Laboratory of Environmental Geochemistry, Institute of Geochemistry, Chinese Academy of Sciences, Guiyang 550002, P.R. China

[3]Soil and Earth Sciences, Institute of Agriculture and Environment, Massey University, Palmerston North, New Zealand

Correspondence to: X. B. Feng (fengxinbin@vip.skleg.cn) and B. Meng (mengbo@vip.skleg.cn)

**Abstract.**

Rice paddy plantation is the dominant agricultural land use throughout Asia. Rice paddy fields have been identified as important sites for methylmerucry (MeHg) production in the terrestrial ecosystem, and a primary pathway of MeHg exposure to human in mercury (Hg) mining areas. We compared the source and distribution of Hg species in different compartments of the rice paddy during a complete rice-growing season at two different typical Hg-contaminated mining sites in Guizhou province, China: an abandoned site with high Hg concentration in soil but low concentration in the atmosphere, and a current-day artisanal site with low concentration in soil but high concentration in the atmosphere. Our results showed that the flux of new Hg to the ecosystem from irrigation and atmospheric deposition was insignificant relative to the pool of old Hg in soil; the dominant source of MeHg to paddy soil is in situ

methylation of inorganic Hg (IHg). Elevated MeHg concentrations and the high proportion of Hg as MeHg in paddy water and the surface soil layer at the artisanal site demonstrated active Hg methylation at this site only. We propose that the in situ production of MeHg in paddy water and surface soil is dependent on elevated Hg in the atmosphere and the consequential deposition of new Hg into a low pH anoxic geochemical system. The absence of depth-dependent variability in the MeHg concentration in soil cores collected from the abandoned Hg mining site, consistent with the low concentration of Hg in the atmosphere and high pH of the paddy water/irrigation water, suggested that net production of MeHg at this site was limited. We propose that the concentration of Hg in ambient air is an indicator for the risk of MeHg accumulation in paddy rice.

## 1    Introduction

Reports of methyl mercury (MeHg) contamination of rice grain (*Oryza sativa*) have recently focussed scientific attention on this important agricultural crop (Qiu et al., 2008; Zhang et al., 2010a; Meng et al., 2010, 2011, 2014). Numerous studies have reported high MeHg concentrations in rice grain collected from Indonesia (Krisnayanti et al., 2012) and different parts of China (Horvat et al., 2003; Qiu et al., 2008; Meng et al., 2014). The MeHg concentration in rice grain (brown rice) can be enhanced even in cases where soil is not significantly elevated in Hg (Zhang et al., 2010a; Horvat et al., 2003). Meng et al. (2014) specified that the majority (~80 %) of MeHg was found in edible white rice. A common theme to these studies is the presence of a high Hg flux into the environment through mining or other industrial contamination that discharges into the atmosphere.

Rice paddy plantation is one of the most prevalent land uses in South and East Asia where rice is the dominant foodstuff (FAO, 2002). Rice throughout Asia is generally cultivated in paddy soil and this ephemeral wetland is known to be a environment for Hg methylation. Current understanding is that the mobility and methylation of Hg in ephemeral flooded soil is determined by a range of factors, such as redox potential, pH, dissolved organic carbon, sulfur, iron, and dissolved Hg content (e.g., Ullrich et al., 2001; Benoit et al., 2001). Mercury methylation is largely facilitated by a subset of sulfate-reducing bacteria (SRB) (Gilmour et al., 1992) and/or iron-reducing bacteria (Fleming et al., 2006) in anoxic conditions. Specially, the methylation of inorganic Hg (IHg) in paddy soil primarily occurs through a

process mediated by sulfate-reducing bacteria (Peng et al., 2012; Rothenberg and Feng, 2012; Wang et al., 2014; Liu et al., 2014; Liu et al., 2009; Somenahally et al., 2011). MeHg accumulated throughout a rice plant during the growing season can be readily translocated to grain during rice-seed ripening (Meng et al., 2011). Rice paddy fields have therefore been identified as important sources of MeHg in the terrestrial ecosystem (Meng et al., 2010, 2011), and a primary vector for human exposure to MeHg in Hg mining areas (Feng et al., 2008; Zhang et al., 2010b).

The general consensus among Hg researchers is that soil is the principle source of MeHg in rice plants, whereas Hg from the ambient air is the principal source of IHg in rice grain (Meng et al., 2010, 2011, 2012, 2014; Qiu et al., 2012; Yin et al., 2013). Recently, Meng et al. (2010, 2011) suggested that newly deposited Hg is more readily transformed to MeHg and accumulated in rice plants than Hg forms with an extended residence time in mining-contaminated soil. Meng et al. (2010) focused on the Wanshan area of China, a region of both historical large-scale and current small-scale mercury mining and showed that  the MeHg concentration in rice grain collected from an active artisanal Hg mining areas ($32\pm14$ ng g$^{-1}$) was significantly higher than in rice grain collected from an abandoned Hg mining area ($7.0\pm3.2$ ng g$^{-1}$). Such studies on MeHg and rice emphasize that factors which control the biochemical cycling of Hg within rice paddy ecosystems are very complex, and include the concentration and distribution of Hg in ambient air, wet/dry deposition, irrigation water, and the solid and liquid phases of soil. These factors in turn impact the absorption, transportation, and accumulation of Hg in rice plants (Meng et al., 2010, 2011, 2012, 2014; Rothenberg and Feng, 2012; Liu et al., 2012; Wang et al., 2014; Peng et al., 2012).

While the source, distribution, and accumulation of IHg and MeHg in rice plants has been reported, no study has presented results from a systemic survey of the concentration of Hg in the various physiochemical fractions of the rice paddy ecosystem. The biochemical processes that control the cycling of Hg in paddy soils impacted by Hg mining are poorly understood. The objectives of the current study were therefore to 1) investigate the speciation and distribution of Hg in paddy soil, and 2) assess the primary source and mechanism for Hg methylation within a Hg mining area. Documenting Hg cycling in rice paddy ecosystems within Hg mining areas is an important step towards better assessing potential health threats that may be associated with rice cultivation in a Hg-contaminated

environment and is necessary to mitigate the risk of MeHg formation in paddy soils used for rice
cultivation in Hg contaminated areas. Better understanding of the distribution of Hg species in paddy
soils within a Hg mining area is necessary to underpin more reliable risk assessment and appropriate
strategies to remediate contaminated soil.
**2    Materials and methods**
**2.1    Site description**
This study was conducted in the Wanshan Hg mining district (E: $109^o07'\sim109^o24'$, N: $27^o24'\sim27^o38'$),
Guizhou province, Sourthwest China, where historical large-scale Hg smelting combined with current
artisanal Hg smelting activities have resulted in Hg contamination of ambient air, water, soil, sediment,
and biota (Qiu et al., 2005; Li et al., 2008, 2009). Two typical Hg contaminated sites within the
Wanshan Hg mining district were selected for this study: an artisanal Hg mining site (Gouxi); and an
abandoned Hg mining site (Wukeng) (Fig. 1). The sampling sites of Gouxi and Wukeng are situated
within the Wanshan district which experiences subtropical monsoon-type climate with an average
annual rainfall of 1200-1400 mm $y^{-1}$ and a perennial mean temperature of 17 $^o$C. Historical Hg mining
activities in the Wanshan area can be dated back to the Qin Dynasty (221 B.C.) but large-scale mining
activities officially ceased in 2001. Mining activity across Wanshan generated an estimated cumulative
$1.0\times10^8$ tons of calcine and waste rock between 1949 and the 1990s. Recently, illegal artisanal Hg and
small-scale mining activities have been revived due to an increase in the global Hg price and domestic
demand.
The Gouxi artisanal Hg mining site is located to the north of Wanshan town (Fig.1). Small-scale
artisanal smelting was ongoing during the rice growing seasons of 2012 when the samples for the
current study were collected. Mercury is released into the atmosphere during artisanal smelting and is
subsequently deposited onto nearby rice paddy fields through wet and dry deposition. The Wukeng
sampling site is located north east of Wanshan town at an abandoned Hg mining area where large
quantities of calcines were deposited along the river.
Paddy field is the primary agricultural land use at both Gouxi and Wukeng. Field sampling for the
current research focused on two $10\times10$ m plots (one at each site) within rice paddies that were
established according to the following methodology: The rice paddies were flooded on $10^{th}$ May; rice
seedlings (hybrid rice) widely grown throughout Guizhou province were transplanted into the
submerged soil 20 days after flooding (1 plant each $25\times25$ cm area on $1^{st}$ June, defined as Day 0).
Thereafter, the two experimental plots were cultivated during the period $1^{st}$ June through $10^{th}$ September
(100 days) 2012. Standing water (2-8 cm) was maintained above the soil surface (flooded condition)
throughout the growing period, from Day 0 to Day 80. The paddy fields were thereafter drained from
Day 80, prior to harvest between Days 90 to 100. During the 10 day draining period, approximately 2-4
cm depth of water was maintained above the soil surface. The paddy plots received water through
precipitation and stream water irrigation, while evaporation to air and seepage to the subsoil were the
primary vectors for water loss. There was no direct runoff from either paddy.
**2.2    Sample collection and preparation**
Five consecutive sampling campaigns were conducted during the rice growing season ($1^{st}$ June-$10^{th}$
September, 2012). The first sampling was initiated 20 days after the plants were planted out ($20^{th}$ June,
2012; Day 20), and thereafter samples were collected on Days 40, 60, 80, and 100 (Day 100 was $10^{th}$
September, 2012; final harvest). The Hg concentration in ambient air was measured at each sampling
time, and samples of cumulative deposition (wet and dry), irrigation water, paddy water, and soil cores
were also collected each time. The relative flux of different Hg vectors to the pool of soil Hg was
subsequently estimated. It should be noted that current study focused on the speciation and distribution
of Hg in the paddy soil during the rice growing season. Rice plant samples were not, however, collected
as part of this study. The paddy fields were dry from Day 90, and therefore irrigation water, paddy water,
and soil pore water samples were not collected on Day 100.
**2.2.1    Mercury in ambient air and wet/dry deposition**
The concentration of total gaseous mercury (TGM, $Hg^0$) in ambient air at both Gouxi and Wukeng was
measured in the field at each sampling time using an automated Hg vapor analyzer (LUMEX, RA-915$^+$,
Ohio Lumex Co., Twinsburg, OH) with detection limit of 2 ng m$^{-3}$. The average Hg$^0$ concentration
during a 10 s interval was quantified and stored in a portable computer. Measurements were carried out
continuously for at least one hour. For each sampling interval 360 data points were collected at each
location.
An integrated bulk precipitation sampler based on the design of Guo et al. (2008) was used in the field
to quantify the concentration of Hg in cumulative precipitation (Oslo and Paris, 1998). Both dry and wet
atmospheric deposition were collected concurrently with the TGM measurement once every 20 days
using this sampling method. Samples collected at each site were poured into two 100 mL pre-cleaned
borosilicate glass bottles for direct and unfiltered total Hg (HgT$_{unf}$) and total MeHg (MeHg$_{unf}$) analysis.
Filtered samples were collected on site using a 0.45 μm disposable polycarbonate filter unit (Millipore),
and subsequently analyzed for dissolved total Hg concentrations (HgT$_f$) and dissolved MeHg
concentration (MeHg$_f$)

### 2.2.2   Irrigation water and paddy water

Samples of irrigation water at both Gouxi and Wukeng were collected at rice paddy inlets on Days 20,
40, 60, and 80. All water samples were collected by hand using ultra-clean handling protocols and
stored in acid-cleaned borosilicate glass bottles. Each bottle was rinsed three times with irrigation water
on site before sample collection. Filtered samples were collected on site through a 0.45 μm disposable
nitrocellulose filter unit (Millipore) HgT$_f$ and MeHg$_f$ analysis. In addition, unfiltered irrigation water
samples were siphoned into pre-cleaned borosilicate glass bottles using a disposable syringe and
analyzed for total Hg (HgT$_{unf}$) and total MeHg (MeHg$_{unf}$).
Paddy water (overlying water) and corresponding soil pore water samples at both Wukeng and Gouxi
were collected at the centre of the two plots on Days 20, 40, 60, and 80, simultaneously to the irrigation
water collection. Firstly, an undisturbed soil core was collected at each sampling site by pushing a pre-
cleaned 6cm diameter polycarbonate core tube into the soil to approximately 20 cm depth. The paddy
water (0-8 cm above the soil surface) in the core tube was siphoned into a 200 ml pre-cleaned
borosilicate glass bottles. One aliquot of the paddy water was then filtered into a 100 ml pre-cleaned
borosilicate glass bottle using a 0.45 μm disposable polycarbonate filter unit (Millipore), and
subsequently analyzed for $HgT_f$ and $MeHg_f$. A second aliquot of paddy water for $HgT_{unf}$ and $MeHg_{unf}$
analysis was immediately transferred into another 100 ml pre-cleaned borosilicate glass bottle.
General Water Quality Characteristics of irrigation water and paddy waterincluding pH, dissolved
oxygen (DO) concentration, and temperature (T) were measured *in situ* using a portable analyzer. All
water samples were promptly acidified on site to 0.5 % (*v/v*) using adequate volumes of ultra-pure
concentrated hydrochloric acid (HCl). The sample bottles were then capped, sealed with Parafilm®,
double-bagged, transported to the laboratory in an ice-cooled container to the lab within 24 h. Prior to
Hg analysis, samples were stored in a refrigerator at +4 ℃ in the dark.
**2.2.3    Soil pore water (liquid phase) and soil core (liquid phase+solid phase)**
The soil cores were immediately sliced on site into 2 cm intervals using a plastic cutter in an oxygen-
free glove box under argon. Firstly, the air (oxygen) in the glove bag was eliminated manually. Then,
the pure argon from a portable argon tank was injected into the glove bag through a Teflon tubing. The
soil samples were placed in acid-cleaned 50-ml plastic centrifuge tubes, capped and sealed with
Parafilm®. All samples were transported in an ice-cooled container to the lab within 24 h and stored at
3-4 °C for further laboratory processes. Following centrifugation (30 min, 3000 r $min^{-1}$, and 5 °C), the
samples were returned to the glove box where the pore water was then filtered through 0.45 μm
disposable nitrocellulose filter unit (Millipore). The filtrate was stored in borosilicate glass bottles and
divided for $HgT_f$ and $MeHg_f$ analysis. The water content of soil cores was estimated by weight loss.
At each sampling time (Days 0, 20, 40, 60, and 80) a second soil core was collected and immediately
placed into liquid nitrogen. This second set of soil cores was transported in a liquid nitrogen-iced
container to the lab within 24 h and then sliced at 2 cm intervals. The sliced soil cores were then freeze-
dried, prior to homogenisation to 200 mesh with a mortar and pestle for analysis of total Hg (THg) and
MeHg. The concentration of each Hg species in this second set of soil cores is therefore the sum of both
liquid and solid phase. Precautions were taken to avoid cross-contamination during sample processing;
the mortar and pestle were thoroughly cleaned after each sample with drinking water followed by
deionized water rinses. The powdered samples were subsequently packed into plastic dishes, sealed in
polyethylene bags and stored in a refrigerator within desiccators for subsequent laboratory analysis.

## 2.3 Sample analysis

All reagents used in this study were of guaranteed quality purchased from Shanghai Chemicals Co. (Shanghai, China).

### 2.3.1 Total Hg and MeHg in soil samples

For THg analysis, a soil sample (0.1-0.2 g) was digested using a fresh mixture of HCl and $HNO_3$ (1:3, $v/v$). THg was determined by cold vapor atomic fluorescence spectrometry (CVAFS, Tekran 2500, Tekran Instruments) preceded by BrCl oxidation, $SnCl_2$ reduction, pre-concentration, and thermo-reduction to $Hg^0$ (U.S. EPA, 2002).

For MeHg analysis, a soil sample (0.3-0.4 g) was prepared using the $CuSO_4$-methanol/solvent extraction (Liang et al., 1996). MeHg in samples was extracted with methylene chloride, then back-extracted from the solvent phase into an aqueous ethyl phase. The ethyl analog of MeHg, methylethylHg ($CH_3CH_3CH_2Hg$), was separated from solution by purging with $N_2$ onto a Tenax trap. The trapped $CH_3CH_3CH_2Hg$ was then thermally desorbed, separated from other Hg species by an isothermal gas chromatography (GC) column, decomposed to $Hg^0$ in a pyrolytic decomposition column (800°C), and analyzed by CVAFS (Brooks Rand Model III, Brooks Rand Labs, U.S.A.) following EPA method 1630 (U.S. EPA, 2001).

### 2.3.2 Total Hg and MeHg in water samples

The analysis of Hg species in water samples was conducted within three weeks of sampling. The $HgT_{unf}$ and $HgT_f$ concentration in water samples was quantified using dual amalgamation CVAFS (Tekran 2500, Tekran Inc., Toronto, Ontario, Canada) following approved methodology (U.S. EPA, 2002). Samples for $HgT_{unf}$ and $HgT_f$ analysis were oxidized with 0.5 % ($v/v$) BrCl (bromine chloride). Excess BrCl was reduced with hydroxyl-ammonium chloride before adding $SnCl_2$ (stannous chloride) to convert $Hg^{2+}$ to volatile $Hg^0$. The $Hg^0$ was trapped by gold amalgamation (U.S. EPA, 2002). Water samples were analyzed for MeHg using CVAFS (Brooks Rand Model III, Brooks Rand Labs, Seattle, WA, USA) following distillation, aqueous phase ethylation, and isothermal GC separation(U.S. EPA, 2001).

## 2.4 Quality control

Quantification for THg and MeHg in soil and water samples was conducted using daily calibration curves with the coefficient of variation ($r^2$) $\geq$ 0.99. Quality control and assurance measurements for all analytes were performed using triplicates, method detection limits, field blanks, matrix spike recoveries, and certified reference materials. Field blanks of water samples were 0.12 ng $L^{-1}$ and 0.011 ng $L^{-1}$ for THg and MeHg, respectively. The method detection limits ($3 \times \sigma$) were 0.02 $\mu$g $kg^{-1}$ for THg and 0.002 $\mu$g $kg^{-1}$ for MeHg in soil samples; 0.02 ng $L^{-1}$ for THg and 0.01 ng $L^{-1}$ for MeHg in water samples, respectively. The variability between the triplicate samples was less than 7.5 % for THg and MeHg analysis for both water and soil samples. Recoveries for matrix spikes in water samples ranged from 88 to 108 % for THg analysis, and from 86 to 113 % for MeHg. The following certified reference materials were employed: Montana soil (SRM-2710, National Institute of Standards and Technology), Loamy Sand 1 (CRM024-050, Resource Technology Corporation), Sandy Loam 3 (CRM021-100, Resource Technology Corporation), and Sediment (BCR-580, Institute for Reference Materials and Measurements). The results of the certified reference material analysis are shown in Table 1.

Statistical analysis was performed using SPSS 13.0 software (SPSS). Mercury concentrations in samples are described by the analysed mean $\pm$ standard deviation (SD) unless otherwise stated. Relationships between covariant sets of data were subjected to regression analysis. Correlation coefficients ($r$) and significance probabilities ($p$) were computed for the linear regression fits. Differences are declared significant for $p < 0.05$. Kolmogorov-Smirnov (K-S) and Kruskal-Wallis (K-W) tests were processed for comparing the differences between the two or more independent datasets (non-parametric tests).

## 3 Results and Discussion

### 3.1 Mercury in ambient air and precipitation

The average TGM concentration in ambient air over the 100 day rice season at Gouxi ($403 \pm 399$ ng $m^{-3}$) was significantly higher than that at Wukeng ($28 \pm 13$ ng $m^{-3}$) and the regional background ($6.2 \pm 3.0$ ng $m^{-3}$) (Table 2). Serious Hg contamination of air was therefore observed at Gouxi during the monitoring

period. The elevated TGM concentration in ambient air at Gouxi compared to Wukeng and the regional
background area (Huaxi) is attributed to the emission of gaseous $Hg^0$ from nearby artisanal Hg smelters
(Meng et al., 2010; Li et al., 2008, 2009).
During the rice growing season, the $HgT_{unf}$ concentration in precipitation (wet and dry deposition) at
Gouxi was elevated (mean=2599±1874 ng $L^{-1}$), and 1-3 orders of magnitude higher than that recorded
for Wukeng (mean=445±296 ng $L^{-1}$) and the regional background measured at Huaxi (mean=27±17 ng
$L^{-1}$) (Table 2). The relative concentration of Hg in precipitation between the three sites was comparable
to the concentration of Hg in the ambient air suggesting that elevated Hg in precipitation at Gouxi can
be linked to the ongoing Hg smelting activities. Mercury in precipitation is therefore a function of the
enhanced flux of both dry and wet deposition of Hg from the atmosphere.
The $MeHg_{unf}$ concentration in precipitation collected from the two sites (Gouxi; 0.48±0.20 ng $L^{-1}$;
Wukeng: 0.30±0.15 ng $L^{-1}$) was similar to the regional background concentration of MeHg (0.28±0.14
ng $L^{-1}$) (Table 2). Furthermore, there was no difference in $MeHg_{unf}$ concentration between the two
sampling sites during the rice growing season (*K-S* test, *p*>0.05). These results confirm previous
suggestions that atmospheric deposition is responsible for the flux of inorganic Hg but not MeHg to
mining areas where artisanal Hg mining is ongoing (Meng et al., 2011).
**3.2    Mercury in irrigation water and paddy water**
The concentration of Hg in irrigation water and paddy water across the two sampling sites is presented
in Table 3. Paddy fields selected in this study were irrigated with local stream water with a high
concentration of Hg due to contamination of streams with calcines and tailings. During the rice growing
season, irrigation water at Wukeng had a significantly higher $HgT_{unf}$ (513±215 ng $L^{-1}$) and $MeHg_{unf}$
(1.7±1.1 ng $L^{-1}$) concentration than at Gouxi ($HgT_{unf}$=159±67 ng $L^{-1}$; $MeHg_{unf}$=0.75±0.65 ng $L^{-1}$).
Mercury concentrations in irrigation water at both sites were significantly higher than the regional
background (*p*<0.05).
Clear differences were observed between the two sites with regard to MeHg concentration and the ratio
of $MeHg_{unf}/HgT_{unf}$ in paddy water. The highest values of $MeHg_{unf}$ in paddy water were all observed at
Gouxi (13±16 ng $L^{-1}$), whereas samples from Wukeng (1.1±0.52 ng $L^{-1}$) maintained a relatively low
MeHg concentration in paddy water throughout the rice growing season. The ratio of MeHg to total Hg
is recognized as a measure of Hg methylation efficiency (Sunderland et al., 2006). In our study, the
$MeHg_{unf}/HgT_{unf}$ ratio was up to 11 % ($MeHg_{unf}/HgT_{unf}$) for paddy water at Gouxi and the mean ratio for
this water compartment was significantly higher than for irrigation water ($0.71 \pm 0.93$ ％) and
precipitation ($0.031 \pm 0.028$ ％) (Table 2 and Table 3). However, there was no significant difference
between the $MeHg_{unf}/HgT_{unf}$ ratios for the various water compartments at Wukeng (K-W test, $p>0.05$).
These results imply active net Hg methylation in paddy fields at Gouxi but not at Wukeng. The
$MeHg_{unf}/HgT_{unf}$ ratios for precipitation ($0.76 \pm 0.41$ %), irrigation water ($2.2 \pm 0.98$ %), and paddy water
($10 \pm 7.9$ %) for both mining sites were elevated relative to the regional background, and we believe this
is due to the lower $HgT_{unf}$ concentration reported for the regional background (Table 2).
**3.3     Mercury in soil profiles**
**3.3.1     Hg in soil pore water**
The vertical distribution of $HgT_f$ and $MeHg_f$ in pore water was monitored over four successive time
intervals during the rice growing season (Fig. 2). The distribution of $HgT_f$ in pore water as a function of
depth was different to that for $MeHg_f$ at both sampling sites. The mean $HgT_f$ concentration in pore
water samples over the 100 days rice growing season was $142 \pm 111$ ng L$^{-1}$ (range: 15-460 ng L$^{-1}$) at
Gouxi and $180 \pm 160$ ng L$^{-1}$ (range: 38-916 ng L$^{-1}$) at Wukeng. The highest concentration of $HgT_f$ in
pore water was measured in the soil surface layer (top 2 cm), and decreased with depth at both sampling
sites. The $HgT_f$ concentration in pore water at Gouxi was relatively constant over time with no
significant difference between the different sampling dates (K-W test, $p>0.05$). At Wukeng, the $HgT_f$
concentration in pore water was time-dependent, with the highest concentration in the surface layer
recorded on Day 20, and the lowest on Day 80 (K-W test, $p<0.01$).
The maximum concentration of $MeHg_f$ in soil pore water (15 ng L$^{-1}$) was observed at Gouxi, and was
approximately double than that at Wukeng (6.6 ng L$^{-1}$). The MeHg concentration in soil pore water
collected at Gouxi was significantly higher than at Wukeng throughout the monitoring periods (K-S test,
$p<0.01$), suggesting different rates of net Hg methylation between the Gouxi and Wukeng sites. The
MeHg$_f$ concentration in pore water was generally highest in the surface soil at Gouxi, and then sharply
declined from a depth of 4 cm. In contrast, the vertical distributions of MeHgT$_f$ in soil pore water of
Wukeng showed little variation, with the exception of small (unexplained) peaks at 10 cm on Day 20
and at 6 cm on Day 60. The proportion of pore water HgT$_f$ that was MeHg$_f$ (MeHg$_f$/HgT$_f$) ranged from
0.50 to 8.7 % (mean value of 2.6±1.7 %) and from 0.089 to 4.8 % (mean value of 1.6±1.1%) at Gouxi
and Wukeng, respectively. Regression analysis revealed a significant and positive correlation between
HgT$_f$ and MeHg$_f$ at Gouxi ($r$=0.75, $p$<0.001, n=40) but not at Wukeng ($r$=0.22, $p$=0.17, n=40) (Fig.3),
suggesting a mechanistic relationship between these two Hg species at the artisanal mining site only.
In order to better understand the factors controlling Hg methylation in rice paddy soil, the concentration
of $Fe^{2+}$, $Fe^{3+}$, $S^{2-}$, and $SO_4^{2-}$ in soil pore water was determined and this data is described, in detail, in a
companion paper (Zhao et al., 2016). Briefly, no discernible vertical trend in $Fe^{3+}$ distribution was
observed in the soil pore water across the two sampling sites during the sampling period. The $Fe^{2+}$
concentrations in soil pore water at Gouxi exhibited a narrow range (41~417 μM), relative to that at
Wukeng (2.3~843 μM). The $S^{2-}$ concentration in the soil pore water showed limited variation with depth
at Wukeng (mean=0.70±0.36 μM, range=0.07~1.2 μM) relative to Gouxi (mean=1.8±0.79 μM,
range=0.69~3.8 μM), with the highest value recorded in the surface soil layer at both sites. Temporal
variation of sulfide concentrations at Wukeng and Gouxi was insignificant (K-W test, p=0.73 and
p=0.33 for Wukeng and Gouxi, respectively). The highest $SO_4^{2-}$ concentrations were recorded in the
surface soil layer and decreased with depth across the two sampling sites. As described in the
companion paper (Zhao et al., 2016), $SO_4^{2-}$ stimulation of SRB activity was a potentially important
metabolic pathway for Hg methylation in the rice paddy soil at the two Hg mining sites, while iron
cycling in the rice paddies could impact the availability of Hg in pore water for methylation.
**3.3.2   Mercury in soil cores**
The concentration and distribution of THg as a function of depth in soil cores at Gouxi and Wukeng is
shown in Fig. 4. Over the rice growing season, the mean concentration of THg in soil was 3.2±0.75 mg
kg$^{-1}$ (0.88–4.4 mg kg$^{-1}$) and 38±4.8 mg kg$^{-1}$ (27–48 mg kg$^{-1}$) at Gouxi and Wukeng, respectively. The
THg concentration in paddy soil collected from both Wukeng and Gouxi was higher than the domestic

environmental quality standard for paddy fields in China (0.5 mg kg$^{-1}$) (GB15618-2008), and considered non-suitable for agricultural or residential use according to the level III criterion (1.5 mg kg$^{-1}$) in the Chinese national standard for soil environmental quality. The THg concentration in soil cores showed no significant difference with depth although there was a nominal trend towards decreasing concentration at Gouxi. For all depths the THg concentration in soil was elevated at Wukeng relative to Gouxi, reflecting a greater degree of historical contamination at Wukeng due to a long period of commercial mining activities.

In contrast to THg, the MeHg concentration in soil cores showed significant variation with depth and time (Fig. 4). The MeHg concentration in soil cores at Gouxi showed a maximum value at the water-soil interface and decreased with depth on sampling Days 20 to 80. On Day 100, however, there was no increased MeHg concentration at the surface. The MeHg concentration in Wukeng soil cores showed very little variation with depth, and the MeHg concentration at this site, for all depths, was significantly lower than at Gouxi (K-S test, $p<0.001$). Measured MeHg concentrations at the top of the Gouxi soil profile varied from 0.76 ng g$^{-1}$ to 6.2 ng g$^{-1}$, but remained relatively stable at Wukeng (range: 0.80–3.8 ng g$^{-1}$). Comparison of the MeHg concentration and distribution patterns between the two sites supports the hypothesis of active Hg methylation in the Gouxi soil only.

Methylation can be affected by the pH and organic matter content of soil, and an analysis of soil physiochemical parameters in the soil cores of this study is reported in a companion paper (Zhao et al., 2016). Briefly, the mean organic matter in soil cores was 4.8±0.75 % and 3.5±0.59 % at Gouxi and Wukeng, respectively. The mean soil pH  was the same for both sites (6.7±0.10 at Gouxi and 6.6±0.14 at Wukeng) and did not change as a function of sampling time, despite the  variation reported for irrigation water and paddy water at Wukeng  in the current study (Table 5). The consistency of soil pH throughout the sampling period indicates that irrigation water and paddy water have little influence on bulk soil pH. Statistical analysis revealed that there is no direct impact of pH and organic matter content on the MeHg concentration in soil across the two sampling sites, indicating that absolute pH and organic matter might not be the most important factors regulating Hg methylation activity (Zhao et al., 2016).

Changing redox parameters over the rice growing season may affect the process of Hg methylation. Previous studies have observed that in artificially Hg-polluted soil, Hg bioavailability for methylation can be significantly affected by the level of water saturation (Rothenberg and Feng, 2012; Wang et al., 2014; Peng et al., 2012). Peng et al. (2012) specified that intermittent flooding, as opposed to continuous flooding, could reduce soluble Hg concentrations and inhibit Hg methylation in the rice rhizosphere, subsequently decreasing the accumulation of MeHg in rice grain. Flooded conditions enhance anaerobic microbial activities and increase MeHg yields. The drying of a paddy field is an important cultivation step to control rice plant tillering and increase yield. Therefore, one possible reason for the considerably elevated MeHg concentrations in soil at Gouxi between Day 20 and Day 80 relative to Day 100 is an enhancement of Hg bioavailability and numbers of SRB under flooded conditions that stimulated Hg methylation, and increased the soil MeHg concentration (Wang et al., 2014). As the paddy field dried from Day 80, some degree of net MeHg degradation may have occurred, which could be attributed to the decreased SRB numbers and proportion of Hg methylators in the rhizosphere under aerobic conditions (Wang et al., 2014). This could have contributed to a decreasing trend in soil MeHg concentration during the harvest period.

The profile of MeHg concentration with depth at Wukeng indicates limited MeHg production in this soil despite a significantly higher THg concentration when compared with Gouxi. The average concentration of THg in soil cores collected from Gouxi was 1 order of magnitude lower that at Wukeng, whereas the MeHg concentration in soil cores at Gouxi was significantly higher than at Wukeng (K-S test, $p<0.001$) during the rice growing season. Further comparison reveals that the average MeHg concentration in the surface soil layer (2 cm) at Gouxi was approximately 3 times higher than that at Wukeng. To explain this apparent anomaly, differences in the source and pool of Hg at each site need to be considered.

### 3.4 The relative mercury flux of different vectors to the soil Hg pool

In order to access the input of the various sources of Hg to paddy fields, we estimated the relative flux of different Hg vectors (atmospheric- and irrigation-derived Hg) to the rice paddy soil during the rice growing season. The hypothesis of our study, established from existing literature and data we have

collected, is that the MeHg concentration in Gouxi soil will be greater than Wukeng due to the a higher flux of IHg in the atmosphere at Gouxi through current-day Hg smelting. Furthermore, we believe that the MeHg concentration will be greater in surface soil, as this is the receiving environment for freshly deposited IHg. To avoid confusion in the naming of the various Hg pools, we refer to deposited Hg as 'new Hg'. Mercury present in the soil is termed 'old Hg', which can be either of geogenic and anthropogenic origin. The current study did not attempt to distinguish between these two sources of old Hg.

Two key assumptions have been adopted in the following discussion: 1) that during the flooded period of the rice growing season, the depth of overlying water remained the same, i.e. that an equilibrium existed between irrigation and water loss; and 2) that all Hg species derived from deposition or irrigation entered into the paddy soil and there was no loss. Using these assumptions we derived the following equations to quantify the relative flux of different sources of Hg to the rice paddy soil based on the measured Hg concentration in atmospheric deposition, irrigation water, and soil:

$$F_p = M_p \times A \times C_p \times 10^{-3} \tag{1}$$

$$F_w = M_W \times A \times C_w \times 10^{-3} \tag{2}$$

$$M_S = C_S \times W \tag{3}$$

where, $R_p$ and $R_w$ are the relative flux of deposition and irrigation to the the rice paddy soil, respectively; $M_S$ is the amount of old Hg species present in the paddy soil; $M_p$ is cumulative rainfall during the rice growing season (17.76 cm); $M_w$ is the cumulative amount of irrigation water during the rice growing season (cm); $C_p$ and $C_w$ are the concentration of Hg (e.g. $HgT_{unf}$ and $MeHg_{unf}$) in deposition and irrigation water (ng $L^{-1}$); $C_s$ is the concentration of Hg species (e.g. THg and MeHg) in the soil cores during the rice growing season (ng $g^{-1}$); W is the mass of the specific soil cores (g); and A is the cross-sectional area of the soil core ($cm^2$).

The amount of the irrigation water during the rice growing season can be calculated using the following equation (Lan et al., 2010):

$$M_w + M_p = M_e + M_i + M_t + M_o \tag{4}$$

where, $M_e$ (cm) is the cumulative amount of water lost by evaporation; $M_i$ (cm) is the cumulative amount of water lost by infiltration; $M_t$ (cm) is the cumulative amount of water lost water by transpiration; $M_o$ is the cumulative amount of water lost by other pathways (e.g. animal activities and draining the paddy during the ripening period). According to published literature, values for $M_e$, $M_i$, $M_t$, and $M_o$ specific to the study area are 40 cm, 0.45 cm, 1.4 cm, and 0.38 cm, respectively (Lan et al., 2010).

Using Equations 1-2, the relative flux of the different sources of Hg (THg and MeHg) to the rice paddy soil during the rice growing season was calculated. Furthermore, the amount of native THg and MeHg present in the paddy soil (20 cm depth) was calculated using Equation 3. The calculated data are listed in Table 4. The calculations showed that the MeHg flux to the rice paddy soil attributable to atmospheric deposition (Gouxi=3.3 mg ha$^{-1}$; Wukeng=2.1 mg ha$^{-1}$) and irrigation (Gouxi=1.8 mg ha$^{-1}$; Wukeng=4.2 mg ha$^{-1}$) was 3 orders of magnitude smaller than the amount of native MeHg already present in the paddy soil (Gouxi=2026 mg ha$^{-1}$, Wukeng=1613 mg ha$^{-1}$). A similarly low value for atmospheric deposition (Gouxi=$1.8 \times 10^{-2}$ mg ha$^{-1}$; Wukeng=$3.1 \times 10^{-3}$ mg ha$^{-1}$) and irrigation water (Gouxi=$0.39 \times 10^{-3}$ mg ha$^{-1}$; Wukeng=$1.3 \times 10^{-3}$ mg ha$^{-1}$) flux was apparent for THg (Table 4) when compared with the soil THg pool (Gouxi=3.2 mg ha$^{-1}$; Wukeng=32 mg ha$^{-1}$). Our calculations therefore suggest that despite the highly elevated THg concentration in atmospheric deposition and irrigation water, the flux of new Hg (MeHg and THg) from external sources was small because of the relatively large pool of old Hg in soil (Dai et al., 2013). Therefore, we propose that the dominant source of MeHg to the paddy soil is in situ methylation of inorganic Hg.

Statistical analysis showed that the THg flux from atmospheric deposition was significantly higher than from irrigation across the two sampling sites (K-S test, $p<0.001$). Furthermore, the THg atmospheric deposition flux at Gouxi was approximately 6 times higher than at Wukeng during the rice growing season. Therefore, we propose that the flux of THg to paddy soil at Gouxi was primarily due to atmospheric deposition associated with ongoing artisanal Hg activities, in agreement with the hypothesis of our study.

**3.5 Source and mechanism for Hg transformation in paddy field**
The mean concentration of $HgT_f$ in paddy water at Wukeng ($197\pm78$ ng $L^{-1}$) was proximately 2 times
higher than that at Gouxi ($105\pm58$ ng $L^{-1}$), whereas the $MeHg_f$ concentration in paddy water at Gouxi
($4.7\pm4.2$ ng $L^{-1}$) was approximately 8 times higher than that at Wukeng ($0.62\pm0.29$ ng $L^{-1}$) (Table 3).
Furthermore, the concentration of $MeHg_f$ in paddy water at Wukeng ($0.62\pm0.29$ ng $L^{-1}$) was
significantly higher than that in precipitation ($0.14\pm0.07$ ng $L^{-1}$), but significantly lower than in
irrigation water ($0.96\pm0.50$ ng $L^{-1}$) and soil pore water ($1.7\pm0.88$ ng $L^{-1}$) in the soil surface layer during
the rice growing season (K-S test, $p<0.001$) (Table 2 and Table 3). Generally, there are three possible
sources of MeHg in the paddy water: 1) in situ production being controlled by chemistry condition (e.g.
redox and pH), 2) diffusion of MeHg from underlying soil, and 3) MeHg flux of atmospheric deposition
and irrigation. We propose that baseline $MeHg_f$ in paddy water at Wukeng is primarily due to the
diffusion of MeHg from the surface layer of sediment and MeHg flux from atmospheric deposition and
irrigation.
The sampling site for the Wukeng paddy was located next to a calcine pile and the proximity of this
waste had a major impact on water chemistry. Both the irrigation water (pH=$11\pm0.45$) and paddy water
(pH=$8.6\pm1.3$) were alkaline during the rice growing season (Table 5). We suggest that the alkaline
conditions of the irrigation at Wukeng could restrain Hg methylation and/or stimulate MeHg
demethylation in paddy water (Ullrich et al., 2001). Rothenberg et al. (2012) reported that alkaline
paddy water (pH >11) at highly-contaminated mining sites can restrain the bioavailability of $Hg^{2+}$ for
Hg methylation, resulting in lower pore water and soil MeHg concentrations despite higher total Hg
concentrations. The findings of our study are in agreement with those of Rothenberg et al. (2012).
In contrast, the $MeHg_f$ concentration in paddy water at Gouxi ($4.7\pm4.2$ ng $L^{-1}$) was significantly higher
than in precipitation ($0.33\pm0.17$ ng $L^{-1}$) and irrigation water ($0.31\pm0.30$ ng $L^{-1}$), but significantly lower
than in soil pore water ($7.8\pm5.2$ ng $L^{-1}$) in the soil surface layer during the rice growing season (K-S test,
$p<0.001$) (Table 2 and Table 3), with the data at Day 80 as an exception. The maximum $MeHg_f$
concentration was not recorded for the surface soil pore water (3.6 ng $L^{-1}$) but for the paddy water (4.7
ng $L^{-1}$) at Day 80. The implication is that MeHg in this region is not only due to MeHg diffusion from
surface soil and/or the MeHg flux of atmospheric precipitation and irrigation, but also from in situ

methylation in anoxic water with relatively low pH (pH=6.9 on Day 80) (Table 5). Gilmour and Henry (1991) specified that low pH and anaerobic condition not only increase methylation rates but also decrease demethylation rates, resulting in net production of MeHg. Both paddy water and irrigation water at Gouxi exhibit pH and redox conditions that can be considered optimal for Hg methylation (Table 5), favouring net methylation in the paddy water (Ullrich et al., 2001). Active Hg methylation within the Gouxi rice paddy is implied in this study. However, data, to directly support this hypothesis are limited. To better understand this observation, further work needs to be done.

During the rice growing season, $HgT_{unf}$ in paddy water exceeded the EPA water-quality criterion of 50 ng $L^{-1}$ (U.S. EPA, 2000). No regulatory criterion for MeHg exists, but Rudd (1995) suggested that MeHg above a concentration of 0.1 ng $L^{-1}$ is elevated and is likely to lead to significant MeHg bioaccumulation. During the rice growing season, photo demethylation can reduce paddy water MeHg. However, the MeHg concentration in both filtered and unfiltered paddy water samples at both sites exceeded 0.1 ng $L^{-1}$ (Table 3), confirming that rice paddies across the Hg mining area are an exposure pathway for MeHg and may have direct implications for human and wildlife health. Previous studies have indicated that vertebrates and fish cultivated in flooded rice paddies will accumulate MeHg to critical threshold levels within 30 days (Ackerman et al., 2010a, 2010b). In rice paddy fields that combine rice and fish cultivation, potential co-exposure of MeHg through rice and fish consumption should receive more attention (Qiu et al., 2008; Feng et al., 2008; Lansing and Kremer, 2011).

Our finding that MeHg concentrations in surface soil at Gouxi were much higher than those at Wukeng indicate that newly deposited mercury can be expected to rapidly methylate after deposition. The peak concentration of MeHg in paddy soil at Gouxi, was at the soil-water interface and decreased with depth. As concluded in a companion paper (Zhao et al., 2016), absolute pH and organic matter might not be the most important factors regulating Hg methylation activity in rice paddy soil. Therefore, we believe that a restricted supply of newly deposited Hg to depths below the soil-water interface is a plausible explanation for the sharply reducing concentration of MeHg with depth at Gouxi; newly deposited Hg is constrained to surface soil and cannot be transferred to lower depth. Therefore, a direct positive relationship between $HgT_f$ and $MeHg_f$ concentrations in soil pore water was observed at Gouxi during the rice growing season (see section 3.3.1).

The Wukeng site has received significant historic Hg deposition as a function of large scale mining, but is not currently receiving significant inputs of fresh Hg. Atmosphere-derived mercury is physically unstable and bioavailable when it first enters the rice paddy (Hintelmann et al., 2002; Schuster, 2011). Immediate reactions of this new Hg with soil constituents are governed by adsorption-desorption interactions with soil surfaces (Schuster, 1991), which favour the retention of Hg in the surface layers of the soil profile. Over time this newly deposited Hg will be transformed into more stable, less available forms (Schuster, 1991), and the net methylation potential of this Hg will consequently decrease. The relatively low MeHg concentration in soil at Wukeng is indicative of old Hg which has become tightly bound to soil complexes over time, and is unavailable for methylation (Hintelmann et al., 2002). Consequently, there is no correlation between $HgT_f$ and $MeHg_f$ in soil pore water at Wukeng (see section 3.3.1). Our data indicates that the THg concentration in soil is not a reliable indicator of Hg methylation potential in soil. Instead, the concentration of bioavailable or new Hg must be considered, in agreement with the findings of Meng et al. (2010, 2011).

### 3.6   Implications of this work to environmental risk assessment

Elevated MeHg concentrations combined with an elevated MeHg% in surface soil active Hg methylation processes are occurring in Gouxi rice paddy soil. The Hg methylation rate is a function of an elevated Hg concentration in atmosphere. The absence of depth-dependent variability in the MeHg concentration in soil cores at Wukeng is consistent with the low concentration of Hg in ambient air and corresponding atmospheric deposition. The in situ production of MeHg in Wukeng soil, despite the elevated concentration of THg, is low. Our results demonstrate that soil is the primary source of MeHg for paddy rice, and we believe that elevated MeHg in rice poses a potential threat to wildlife and local residents. Mercury in surface soil that has been derived from atmospheric deposition is susceptible to methylation in the rice paddy ecosystem immediately after deposition. Consequently, net MeHg production is principally governed by the supply of fresh deposited Hg to soil.

The relationship between MeHg and fresh deposited Hg implies that the concentration of Hg in ambient air could be used as a monitoring tool to assess the relative risk of MeHg production in the rice paddy environment, and the possible risk to human health that may be associated with the accumulation of this

MeHg in rice grain. However, we cannot distinguish between newly deposited Hg and old Hg stored in paddy soil over decades and ongoing research is necessary to continue to develop an improved understanding of Hg dynamics in rice paddy soils. When comparing relative risk between different vectors for Hg contamination (i.e. small-scale or historic large-scale mining), quantification of the pool of Hg available for methylation is critical to estimating relaible methylation rates. Ongoing work is urgently needed to further ascertain the relative importance of newly deposited Hg versus in situ Hg to the bioavailabile pool of Hg that can be methylated in rice paddy ecosystems.

Because MeHg can be demethylated to IHg biotically and abiotically in soil or paddy water, rapid cycling occurs between the IHg and MeHg pools. The current study was limited to the rice growing season, not the entire year or an period of time. Therefore our results define the initial rather than long-term influence of newly deposited Hg on MeHg production. The overall contribution of old versus newly-deposited Hg to the pool of Hg in paddy soil that is available for methylation will likely depend on the balance of Hg deposition and the rate at which this deposited Hg binds to soil constituents, and the magnitude of the IHg flux in the atmosphere. Our study provides no information on the extent to which the MeHg concentration in rice paddies will respond to Hg emission controls which seek to reduce the flux of atmospheric Hg. The response of the paddy ecosystem to reductions in Hg emissions will depend on how long previously deposited Hg has been stored in paddy soil and its availability to SRB. This issue is poorly understood, but previously reported declines in Hg loading suggest that MeHg levels in soil at abandoned Hg mining areas begin to respond within a few years of Hg reductions (Rothenberg et al., 2012). This provides hope that environmental risk mitigation strategies based on a more detailed understanding of the rice-paddy ecosystem at mining contaminated sites can be effectively enacted to protect human health.

**Acknowledgements**

This research was financed by the National Key Basic Research Program of China (973 Program 2013CB430004), financed by natural science foundation of China (41173126, 41573135, 41203091, and 41473123), and also financed by natural science foundation of Guizhou (2012-2333).

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

Table 1. List of certified reference materials used in the present study and corresponding analytical result.

| Producer | CRM | $n$ | Hg speciation | Obtained value | Certified value |
|----------|-----|-----|---------------|----------------|-----------------|
| NIST | SRM-2710 | 10 | THg (mg kg$^{-1}$) | 32.4±0.7 | 32.6±1.8 |
| RTC | CRM024-050 | 10 | THg (mg kg$^{-1}$) | 0.70±0.02 | 0.71 |
| RTC | CRM021-100 | 10 | THg (mg kg$^{-1}$) | 4.73±0.15 | 4.7 |
| IRMM* | BCR-580 | 20 | MeHg (mg kg$^{-1}$) | 0.070±0.007 | 0.075±0.004 |

IRMM: Institute for Reference Materials and Measurements
NIST: National Institute of Standards and Technology.
RTC: Resource Technology Corporation

Table 2. Hg in ambient air and precipitation at artisanal Hg mining site (Gouxi), abandoned Hg mining
site (Wukeng), and regional background of Huaxi (mean±SD)

| Sampling sites | Ambient air[1] | Precipitation[1] | | | | |
|---|---|---|---|---|---|---|
| | $Hg^0$ (ng m$^{-3}$) | $HgT_{unf}$ (ng L$^{-1}$) [3] | $HgT_f$ (ng L$^{-1}$) [3] | $MeHg_{unf}$ (ng L$^{-1}$) [3] | $MeHg_f$ (ng L$^{-1}$) [3] | $MeHg_{unf}/HgT_{unf}$ (%) |
| Gouxi | 403±388 | 2599±1874 | 648±672 | 0.48±0.20 | 0.33±0.17 | 0.031±0.028 |
| Wukeng | 28±13 | 445±296 | 164±166 | 0.30±0.15 | 0.14±0.07 | 0.16±0.20 |
| Huaxi[2] | 6.2±3.0 | 27±17 | | 0.28±0.14 | | 0.76±0.41 |

[1] Hg species concentrations in ambient air and precipitation were averaged with data sets of five sampling campaigns at
Days 20, 40, 60, 80 100.
[2] data were obtained from Zheng, (2007), Meng et al. (2010) and Meng (2011).
[3]$HgT_{unf}$, unfiltered total mercury; $MeHg_{unf}$, unfiltered methylmercury;
Table 3. Hg irrigation water and paddy water at artisanal Hg mining site (Gouxi), abandoned Hg mining site (Wukeng),
and regional background of Huaxi (mean±SD)

| Sampling sites | Irrigation water[1] | | | | | Paddy water[1] | | | | |
|---|---|---|---|---|---|---|---|---|---|---|
| | $HgT_{unf}$ (ng L$^{-1}$) | $HgT_f$ (ng L$^{-1}$)[3] | $MeHg_{unf}$ (ng L$^{-1}$) | $MeHg_f$ (ng L$^{-1}$)[3] | $MeHg_{unf}/HgT_{unf}$ (%) | $HgT_{unf}$ (ng L$^{-1}$) | $HgT_f$ (ng L$^{-1}$) | $MeHg_{unf}$ (ng L$^{-1}$) | $MeHg_f$ (ng L$^{-1}$) | $MeHg_{unf}/HgT_{unf}$ (%) |
| Gouxi | 159±67 | 39±9.4 | 0.75±0.65 | 0.31±0.30 | 0.71±0.93 | 189±117 | 105±58 | 13±16 | 4.7±4.2 | 5.9±4.4 |
| Wukeng | 513±215 | 195±45 | 1.7±1.1 | 0.96±0.50 | 0.45±0.53 | 430±279 | 196±78 | 1.1±0.52 | 0.62±0.29 | 0.48±0.63 |
| Huaxi[2] | 7.1±4.0 | | 0.14±0.044 | | 2.2±0.98 | 7.5±4.3 | | 0.71±0.66 | | 10±7.9 |

[1] Hg species concentrations in irrigation water and paddy water were averaged with data sets of four sampling campaigns at Days 20, 40, 60, 80.
[2] data were obtained from Zheng, (2007), Meng et al. (2010) and Meng (2011).
[3]$HgT_{unf}$, unfiltered total mercury; $HgT_f$, filtered total mercury; $MeHg_{unf}$, unfiltered methylmercury; $MeHg_f$, filtered methylmercury;

1     Table 4. The relative flux of different vectors to the soil Hg pool (20 cm depth) at an artisanal Hg

2     mining site (Gouxi) and abandoned Hg mining site (Wukeng) during the rice growing season.

| Sampling sites | Hg species in soil cores (20 cm) | Irrigation water flux | Atmospheric deposition flux | Native soil Hg pool (%) |
|---|---|---|---|---|
| Gouxi | THg (kg ha$^{-1}$) | $0.39 \times 10^{-3}$ | $1.8 \times 10^{-2}$ | 3.2 |
| | MeHg (mg ha$^{-1}$) | 1.8 | 3.3 | 2026 |
| Wukeng | THg (kg ha$^{-1}$) | $1.3 \times 10^{-3}$ | $3.1 \times 10^{-3}$ | 32 |
| | MeHg (mg ha$^{-1}$) | 4.2 | 2.1 | 1613 |

1    Table 5. Temperature (T), pH, and dissolved oxygen (DO) in irrigation water and paddy water at the

2    artisanal Hg mining site (Gouxi) and abandoned Hg mining site (Wukeng) (mean±SD, range)

| Sampling sites | Irrigation water | | | Paddy water | | |
|---|---|---|---|---|---|---|
| | T ($^{o}$C) | pH | DO (mg L$^{-1}$) | T ($^{o}$C) | pH | DO (mg L$^{-1}$) |
| Gouxi | 24±1.7 | 8.3±0.24 | 7.4±0.43 | 28±4.4 | 7.2±0.24 | 3.0±0.95 |
| | (23-26) | (8.1-8.6) | (6.9-8.0) | (24-33) | (6.9-7.4) | (1.8-3.9) |
| Wukeng | 25±2.1 | 11±0.45 | 7.4±0.56 | 25±2.7 | 8.6±1.3 | 4.4±0.73 |
| | (23-27) | (11-12) | (6.8-8.1) | (23-29) | (7.3-9.8) | (3.6-5.2) |

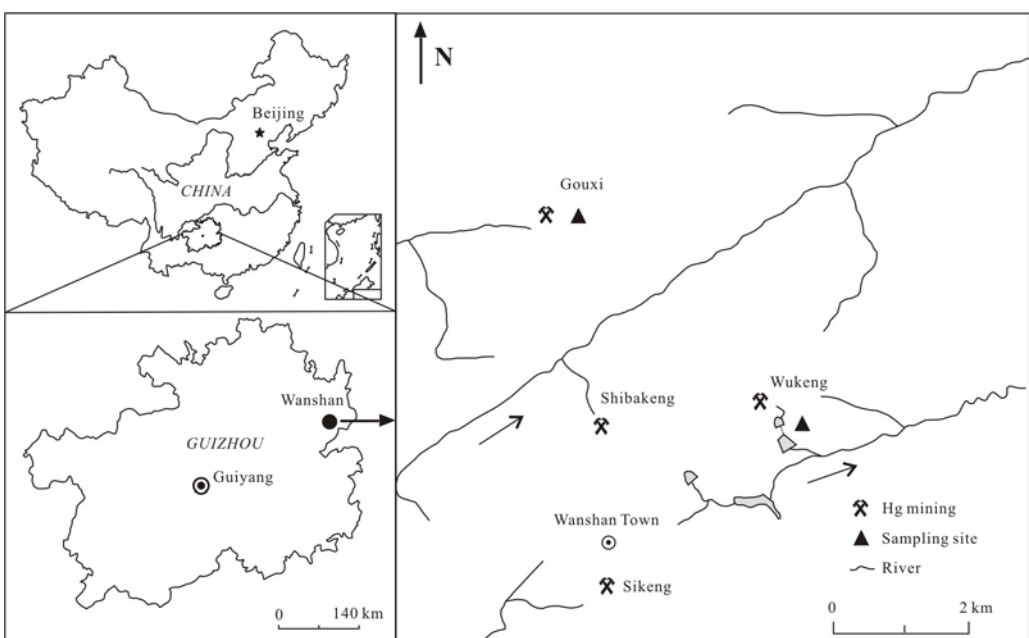

2    Figure 1.Map of the study area and sampling sites including abandoned Hg mining site (Wukeng) and

3    artisanal Hg mining site (Gouxi).

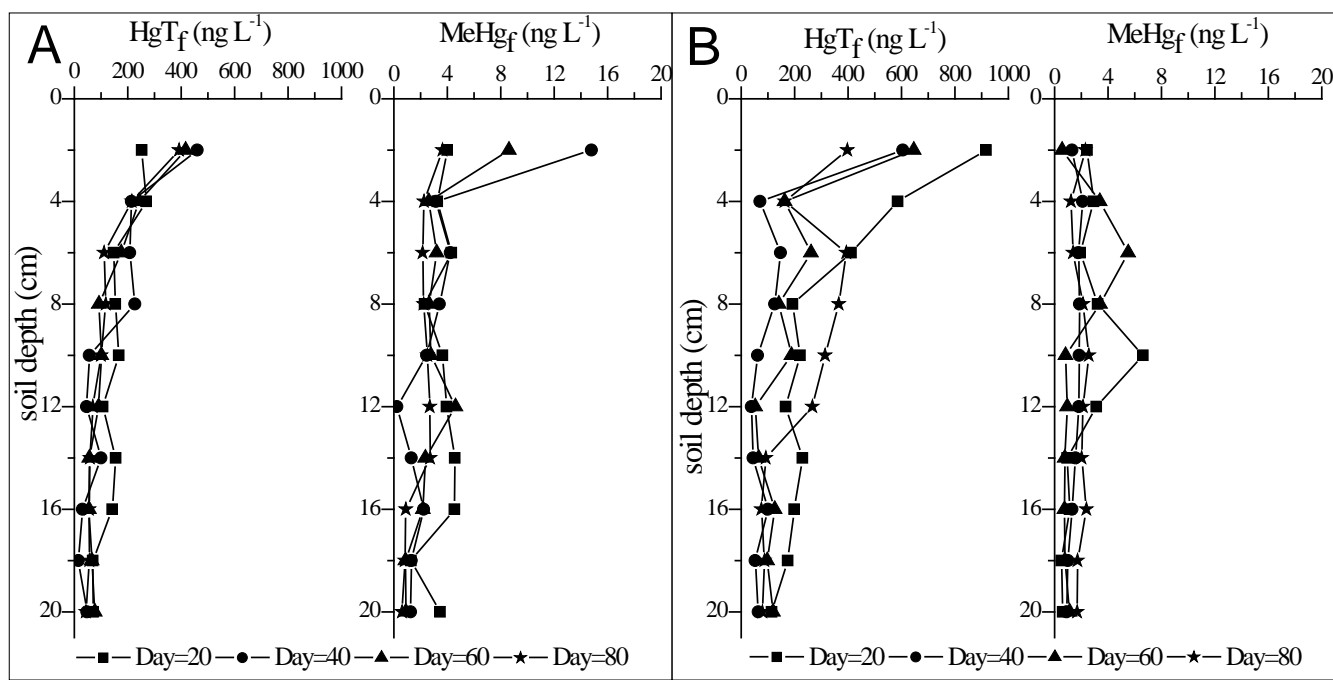

Figure 2. Concentration of $HgT_f$ and $MeHg_f$ (ng $L^{-1}$) in pore water during the rice growing seasonon Days 20, 40, 60, and 80 (A: artisanal Hg mining site of Gouxi; B:abandoned Hg mining site of Wukeng).

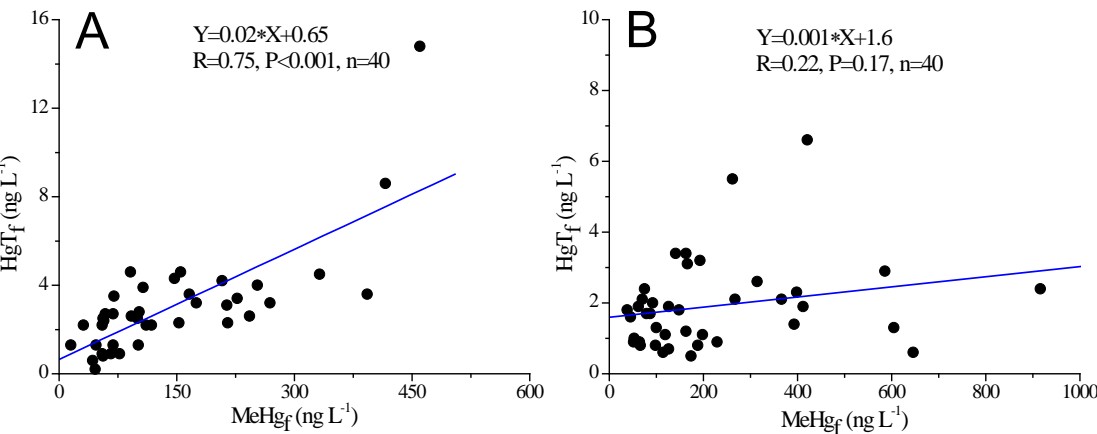

2 Figure 3. Correlation between $HgT_f$ and $MeHg_f$ concentrations in soil porewater during the rice growing

3 season on Days 20, 40, 60, and 80(A: artisanal Hg mining site of Gouxi; B:abandoned Hg mining site of

4 Wukeng).

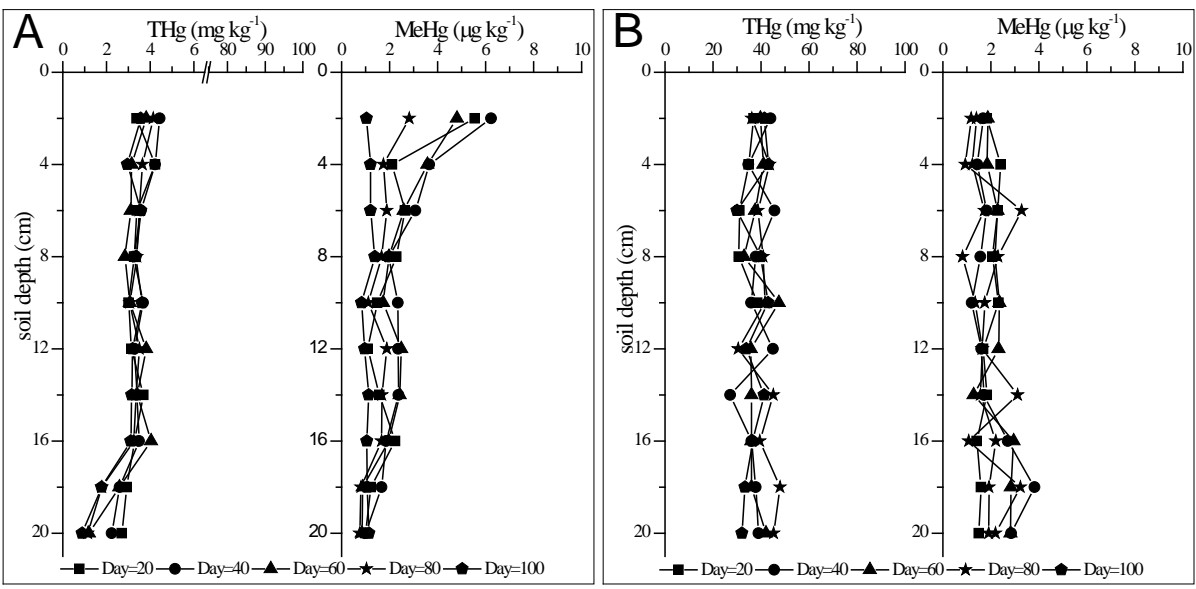

Figure 4. Concentration of THg and MeHg in soil cores during the rice growing seasonon Days 20, 40,

60, 80, and 100 (A: artisanal Hg mining site of Gouxi; B:abandoned Hg mining site of Wukeng).