# Peer review of "Mercury Methylation in paddy soil: Source and distribution of mercury"

_Biogeosciences, 2015_

## Referee Comment (RC1) · Anonymous Referee #1 · 12 Feb 2016

General comments

The study conducted by Zhao et al. takes on a challenging set of important questions regarding mercury in rice paddy soils. The study looks at two different systems, a point source polluted rice paddy at Gouxi, where artisanal mining of Hg occurs, and Wukeng, an abandoned mine with reduced atmospheric deposition of Hg. These two sites are commonly compared to Hauxi, a 'control site' of regional pollution. The design of the study is less-than ideal (see below in specific comments) but the temporal measurements of MeHg makes this study an asset for those interested in Hg in rice paddies.

The manuscript falls short in a few areas. First, the paper lacks a strong, clear state-

ment of hypotheses and the biological/chemical mechanism. By clearly stating the hypotheses, this will greatly aid in organizing the discussion and determine which mechanisms need to be specifically addressed in greater detail in the discussion. From my understanding of the manuscript one hypothesis should be "We hypothesize that the Gouxi site will have greater MeHg than the Wukeng site because greater atmospheric deposition of 'new Hg' is more susceptible to methylation". A second hypothesis could be "We expect greater MeHg in the upper mineral soil horizons at the Gouxi site than Wukeng site because 'old Hg' is less susceptible to methylation". These are just suggestions but explicitly stating them is needed to guide the discussion. In addition to the missing hypotheses, the inclusion of the water-atmospheric model was unnecessary in lieu of a more simplistic comparison of fluxes. The modeling made many assumptions that also made it unreasonable to apply (see specific comments). Lastly, Hg in Oryza sativa data should be presented in this study. Since the uptake of MeHg and Hg by rice is central to this study, linking the belowground processes to the plant would be a great addition to the study.

Specific comments

The introduction lacks discussion of microbial and chemical mechanisms responisible for methylation. Even if you are not testing for methylating bacteria, the mechanisms of methylation should not be absent. In the experimental design, I understand the practicality of monitoring two, 10 X 10 meter rice paddies. However, sampling one plot multiple times to assess an treatment/affect is potentially pseudoreplication. For Section 3.3.2 on Soil Cores, the physical and chemical data of the soil cores should be included in the discussion. Was dissolved oxygen, sulfate, Fe or other important electron acceptors measured and comparable through time? Page 13 Line 1 – 9: The results should be better integrated with existing knowledge about the effect of microbial production of MeHg in flooded soils. The model is an interesting thought-experiment based on a number of assumptions such as negligble amounts of Hg volatilizing from the water surface and dynamic equilibrium of the aqueous solution. However, the as-

sumption I find the hardest to justify is that the system is behaving as an unsaturated soil (as cited in Munthe and Hintelmann) and not behaving as a water-sediment system. I understand it simplifies the system to a traditional unsaturated agricultural system but the fact is the water and saturated soil (now behaving as a sediment) are exchanging with each other rather than acting as one system. Instead of the model in 3.4, is it possible to just compare the atmospheric Hg fluxes, irrigation Hg fluxes, and 'Old Hg' pools and find the same conclusion that Hg was primarily from atmospheric deposition and MeHg is produced in situ? Page 15 Line 25: Could you compare your data on estimated Hg methylation with other rice paddies in Asia to assess how alkaline conditions have slowed or retarded Hg methylation?

Technical comments Page 2 Line 10: a strong bioaccumulator? There is an adjective missing. I might suggest re-writing the sentence. Page 2 Line 15: In which part does the rice uptake Hg? Page 2 Line 27: Although its a common term in Hg literature, please define IHg. Page 3 Line 21 – 25: I feel these details should be in the methods since they describe your actions. Page 4: Is it possible to provide coarse latitude and longitude for the Wanshan mining district in the text? Page 7 Line 3: The phrase 'under argon' is in exact. Please re-phrase with details. Page 7 Line 10: Extra period at beginning of sentence. Page 8 Line 10: correct to EPA method 1630. Page 8 Line 12: Please define HgTunf and HgTf explicitly at first use. Page 8 Line 26-Page 9 Line 1: Please include "respectively" to indicate the relationship between the blank concentrations with THg and MeHg. Page 9 Line 13 – 15: Please mention these are non-parametric tests for those unfamiliar with those tests. Pages 9 – 15: It is conventional for this journal to include a space between numbers and symbols, particularly when expressing the mean and standard deviation. Page 15 Line 6: Was this model used for Hg and MeHg?

---

## Referee Comment (RC2) · Anonymous Referee #2 · 23 Feb 2016

GENERAL COMMENTS

The study by Zhao et al. addresses Hg contamination and mthylation in paddy fields in the province of Guizhou, China. This is a topic of high importance for human and ecosystem health in paddy field areas. The study is relevant for Biogeosciences and falls within the Aims and scope.

The methods are well explained. The results are well presented, although the clarity of the Tables should be improved. The discussion is generally good, but could benefit from more links to existing literature. I have two major comments that should be addressed to improve the paper:

[Figure]

1/ the Hg balance model (eq. 3 and 4). I don't really see the added value of this model. Moreover, some assumptions are very strong (eg rice transpiration amount extremely low), and the athors compare the input of 'fresh' Hg (irrigation and deposition) of 1 year, to the 'old' Hg pool accumulated over the years. Therefore the Hg balance results entirely depend on the number of years during which Hg has accumulated in the surface layer (X years of paddy field irrigation, etc.). I recommend to completely revise the model or simply drop it (unless you can clearly demonstrate what it brings to the discussion and how it supports your conclusions)

2/ the authors should pay more attention and discuss in more details to the biochemical processes affecting Hg methylation. What if the Wukeng soil had had a low pH more favorable to methylation? Would the conlusions of historical vs. artisanal Hg mining still hold? The important pH difference between the two sites prevents any conclusion regarding the impact on methylation of the "type" of Hg available (old at Wukeng vs fresh at Gouxi). If the redox and pH conditions are not good for methylation, it will not occur (whatever the 'type' of Hg present in the soil). I strongly recommend to discuss this (with additional literature references), and reformulate the conclusions taking this into account.

SPECIFIC COMMENTS and TECHNICAL COMMENTS

See attached pdf.

Please also note the supplement to this comment:
http://www.biogeosciences-discuss.net/bg-2015-638/bg-2015-638-RC2-supplement.pdf

―――――――――――――――――

**Supplement:**

[revised manuscript text omitted]

---

## Author Comment (AC1) · 25 Mar 2016

On behalf of my co-authors, I would like to thank the anonymous reviewer for dedicating time to provide comments and criticism. The reviewer raises important issues, which have helped us to improve the manuscript. We have completed a substantial revision and our changes are highlighted in red color in the revised manuscript.

Please find in Supplement the Authors' response and the revised manuscript.

Please also note the supplement to this comment:
http://www.biogeosciences-discuss.net/bg-2015-638/bg-2015-638-AC1-supplement.zip

---

## Author Comment (AC2) · 25 Mar 2016

We are again most thankful to the anonymous reviewer for dedicating their time to provide comments and criticism. The reviewer raises many important issues. We have considered these and made appropriate changes to the text which have definitely improves this manuscript. Revisions are shown in red color in the revised manuscript, with our point-by-point response presented. Please find in Supplement the Authors' response and the revised manuscript.
* * *

---

## Author Comment (AC3) · 25 Mar 2016

**Response to comments from reviewers 2**

**GENERAL COMMENTS**

The study by Zhao et al. addresses Hg contamination and mthylation in paddy fields in the province of Guizhou, China. This is a topic of high importance for human and ecosystem health in paddy field areas. The study is relevant for Biogeosciences and falls within the Aims and scope. The methods are well explained. The results are well presented, although the clarity of the Tables should be improved. The discussion is generally good, but could benefit from more links to existing literature. I have two major comments that should be addressed to improve the paper:

**We are again most thankful to the anonymous reviewer for dedicating their time to provide comments and criticism. The reviewer raises many important issues. We have considered these and made appropriate changes to the text which have definitely improves this manuscript. Revisions are shown in red color in the revised manuscript, with our point-by-point response presented as follows.**

(1) the Hg balance model (eq. 3 and 4). I don't really see the added value of this model. Moreover, some assumptions are very strong (eg rice transpiration amount extremely low), and the athors compare the input of 'fresh' Hg (irrigation and deposition) of 1 year, to the 'old' Hg pool accumulated over the years. Therefore the Hg balance results entirely depend on the number of years during which Hg has accumulated in the surface layer (X years of paddy field irrigation, etc.). I recommend to completely revise the model or simply drop it (unless you can clearly demonstrate what it brings to the discussion and how it supports your conclusions)

**Yes, the authors definitely agree with the reviewer's comments concerning the Hg balance model. Based on the reviewer's comments herein jointly with the suggestion from reviewer #1, we simplified the model in the revised manuscript (see detail in section 3.4).**

**Briefly, we estimated the relative flux of different Hg vectors (atmospheric- and irrigation-derived Hg) to the rice paddy soil (depth of 20 cm) during the rice growing season. Furthermore, the amount of native THg and MeHg present in the paddy soil were calculated as well. Our calculated data showed that the MeHg flux to the rice paddy soil attributable to atmospheric deposition (Gouxi=3.3 mg ha$^{-1}$; Wukeng=2.1 mg ha$^{-1}$) and irrigation (Gouxi=1.8 mg ha$^{-1}$; Wukeng=4.2 mg ha$^{-1}$) was 3 orders of magnitude smaller than the amount of native MeHg already present in the paddy soil (Gouxi=2026 mg ha$^{-1}$, Wukeng=1613 mg ha$^{-1}$). A similar low**

atmospheric deposition (Gouxi=1.8×10$^{-2}$ mg ha$^{-1}$; Wukeng=3.1×10$^{-3}$ mg ha$^{-1}$) and irrigation water (Gouxi=0.39×10$^{-3}$ mg ha$^{-1}$; Wukeng=1.3×10$^{-3}$ mg ha$^{-1}$) fluxes were apparent for THg (Table 4), when compared with the soil THg pool (Gouxi=3.2 mg ha$^{-1}$; Wukeng=32 mg ha$^{-1}$). Furthermore, the THg flux of the atmospheric deposition at Gouxi was approximately 6 times higher than at Wukeng during the rice growing season. Our calculations therefore suggest that despite the highly elevated THg concentration in atmospheric deposition and irrigation water, the flux of new Hg (MeHg and THg) from external sources was small because of the relatively large pool of old Hg in soil (Dai et al., 2013). Therefore, we propose that the dominant source of MeHg to the surface soil layer is *in situ* methylation of inorganic Hg (see page 16 lines 12-26).

(2) the authors should pay more attention and discuss in more details to the biochemical processes affecting Hg methylation. What if the Wukeng soil had had a low pH more favorable to methylation? Would the conlusions of historical vs. artisanal Hg mining still hold? The important pH difference between the two sites prevents any conclusion regarding the impact on methylation of the "type" of Hg available (old at Wukeng vs fresh at Gouxi). If the redox and pH conditions are not good for methylation, it will not occur (whatever the 'type' of Hg present in the soil). I strongly recommend to discuss this (with additional literature references), and reformulate the conclusions taking this into account.

The authors agree with the reviewer's comments herein. Based on the reviewer's comments shown above we re-organized the discussion in section 3.3, section 3.5 and section 3.6 in the revised manuscript as follows:

1) Section 3.3.2:

Main physical and chemical parameters including pH and organic matter content in soil cores, were analyzed in this study and available in a companion paper (Zhao et al., 2016). Briefly, organic matter in soil cores averaged 4.8±0.75% and 3.5±0.59% at Gouxi and Wukeng, respectively. The pH values in soil samples, which averaged 6.7±0.10 at Gouxi and averaged 6.6±0.14 at Wukeng, were nearly neutral during the rice growing season. Although the pH values of the irrigation water and paddy water at Wukeng being significantly higher than those at Gouxi (Table 5), no significant difference of pH levels in soil cores was observed between these two sampling sites throughout the five sampling campaigns, indicating that the irrigation water and paddy water have little influence on the values and distributions of pH in soil cores.

Statistical analysis revealed that there is no direct impact of pH and organic matter content on MeHg concentrations in soil core across the two sampling sites, indicating that absolute pH and organic matter might not be the most important factors regulating Hg methylation activity (Zhao et al., 2016). (page 13 lines 24-27 and page 14 lines 1-8).

Changing redox parameters over the rice growing season may affect the process of Hg methylation. Previous studies have observed that in artificially Hg-polluted soil, Hg bioavailability to methylation can be significantly affected by the level of water saturation (Rothenberg and Feng, 2012; Wang et al., 2014; Peng et al., 2012). Peng et al. (2012) specified that intermittent flooding, as opposed to continuous flooding, could reduce soluble Hg concentrations and inhibit Hg methylation in the rice rhizosphere, subsequently decreasing the accumulation of MeHg in rice grain. Flooded conditions enhance anaerobic microbial activities and increase MeHg yields. Drying of a paddy field is an important cultivation step to control rice plant tillering and increase yield. Therefore, one possible reason for the considerably elevated MeHg concentrations in soil at Gouxi between Day 20 and Day 80 relative to Day 100 is an enhancement of Hg bioavailability and numbers of SRB under flooded conditions that stimulated Hg methylation, and increased the soil MeHg concentration (Wang et al., 2014). As the paddy field dried from Day 80, some degree of net MeHg degradation may have occurred, which could be attributed to the decreased SRB numbers and proportion of Hg methylators in the rhizosphere under aerobic conditions(Wang et al., 2014). This likely contributed to a decreasing trend in soil MeHg concentration during the harvest period. (page 14 lines 9-23)

2) Section 3.5

Both the irrigation water (pH=11±0.45) and paddy water (pH=8.6±1.3) were alkaline during the rice growing season (Table 5). We suggest that the alkaline conditions of the irrigation at Wukeng could restrain Hg methylation and/or stimulate MeHg demethylation (Ullrich et al., 2001). Rothenberg et al. (2012) reported that the more alkaline condition (pH >11) in paddy water at the highly-contaminated site could restrain the bioavailability of $Hg^{2+}$ for Hg methylation, resulting in lower pore water and soil MeHg concentrations despite higher total Hg concentrations, which in agree with our study. The MeHg$_{unf}$ concentration in paddy water at Gouxi was significantly higher than the concentration in either the precipitation or the irrigation water and this implies active Hg methylation within the Gouxi rice paddy. Both

paddy water and irrigation water at Gouxi exhibit a pH that can be considered optimal for Hg methylation (Table 4), favouring net methylation in the paddy fields (Ullrich et al., 2001). (page 17 lines 7-17)

As concluded in a companion paper (Zhao et al., 2016), absolute pH and organic matter might not be the most important factors regulating Hg methylation activity in rice paddy soil. Therefore, we believe that restricted supply of newly deposited Hg to depths below the soil-water interface is a plausible explanation for the sharply reducing concentration of MeHg with depth at Gouxi; newly deposited Hg is constrained to surface soil and cannot be transferred to lower depth. Therefore, a direct positive relationship between $HgT_f$ and $MeHg_f$ concentrations in soil pore water was observed at Gouxi during the rice growing season (see section 3.3.1). However, stimulated MeHg production due to favorable geochemical conditions (e.g. sulfate) at Gouxi cannot be excluded, due to the fact that sulfate stimulating the activity of SRB were a potentially important metabolic pathway for Hg methylation in rice paddy soil in Hg mining area (Zhao et al., 2016). (page 18 lines 4-13)

  3) Section 3.6

We cannot distinguish between newly deposited Hg and old Hg stored in paddy soil over decades; this hypothesis needs further testing to obtain such information regarding Hg dynamics. In comparing across different typical Hg contaminated rice paddy fields, the crucial question regards assessing the pool of Hg available for methylation, which is crucial to estimating realistic and accurate methylation rates. To this point, detailed work is urgent to further ascertain the relative importance of newly deposited Hg versus in situ Hg in contribution to the Hg that is methylated and bioavailability in rice paddy ecosystems. (page 19 lines 14-21)

Because MeHg can be demethylated to IHg biotically and abiotically in soil or paddy water, rapid cycling occurs between the IHg and MeHg pools. Current data were limited to only the rice growing season, not the entire year or a long period of time, and therefore our results represented an initial rather than a long-term influence of newly deposited Hg on the MeHg production. Longer-term responses and a more complex landscape are needed in rice paddy ecosystems. The overall contributions of old versus newly deposited Hg to methylation in paddy soil and the overall activity time to change in atmospheric Hg deposition could likely

depend on the balance of Hg deposition and the rate of deposited Hg bound to soil complexes. This hypothesis does not imply that MeHg concentrations in rice paddies will necessarily recover quickly after implementation of Hg emission controls. The response of MeHg levels in rice paddies to reductions in Hg emissions will depend on how long previously deposited Hg is stored in paddy soil and its availability to SRB. This issue is poorly understood, but previous declines in Hg loading suggest that MeHg levels in soil from abandoned Hg mining areas begin to respond within a few years of Hg reductions. (page 19 lines 22-28 and page 20 lines 1-6)

**SPECIFIC COMMENTS and TECHNICAL COMMENTS**

(1) mention somewhere in the abstract the location (China, and at least the province)

**Yes, the detailed location "Guizhou province China" was added in the revised manuscript (see page 1 lines 18-19)**

(2) use "inorganic Hg" (or define the abbreviation earlier)

**Yes, we defined IHg as inorganic Hg in the revised manuscript. (page2 line 1)**

(3) sentence is unclear "in situ production of MeHg is dependent on elevated IHg in the atmosphere and on the deposition of"... ?

**Yes, we re-organized this sentence in the revised manuscript as "We propose that the in situ production of MeHg is dependent on elevated Hg in the atmosphere and on the newly deposited Hg into a low pH anoxic geochemical system"(see page 2 lines 3-4).**

(4) put the references chronologically.

**Yes we re-organized references as (Horvat et al., 2003; Qiu et al., 2008; Meng et al., 2014) in the revised manuscript (page 2 lines 14-15).**

(5) to support the assumed "consensus", more than one author should be provided.

**Yes, two more references (Qiu et al., 2012; Yin et al., 2013) were added in the revised manuscript (page 3 line 10)**

(6) large scale / small-scale: be consistent with the "-"

**Yes, "large scale" was changed to "large-scale" in the revised manuscript (see page 3 line 13).**

(7) these are all the possible compartments where Hg can be found... this sounds like an "empty" argument or too generic

**The authors agree with the reviewer's comments herein. Therefore, we re-organized this sentence in the revised manuscript as follows: "Studies on MeHg and rice emphasize that factors that control the biochemical cycling of Hg within rice paddy ecosystems are very complex including the concentration and distribution of Hg in ambient air, wet/dry deposition, irrigation water, and solid and liquid phases of soil". (see page 3 lines 23-24)**

(8) confusing. What is "assessing the status of Hg species" ?consider rephrasing ("we analyzed Hg speciation in ...")

**The authors agree with the reviewer's comments herein. We re-worked this sentence in the revised manuscript as follows: "To expand our knowledge of the biochemical processes that affect Hg methylation we analyzed Hg speciation in different compartments of the rice paddy ecosystem." (see page 4 lines 5-6).**

(9) if I'm correct you investigate methylation, not other transformations. Then replace by "Hg methylation"

**Yes, we re-worked this sentence in the revised manuscript as follows "to assess the primary source and mechanism for Hg methylation within paddy soil at a Hg mining area." (see page 4 lines 7-8)**

(10) can you be more precise about what you consider to be seepage and outflow? Is seepage infiltration to the subsoil ? Does outflow mean runoff ?

**Yes, seepage infiltration is to the subsoil and outflow means runoff. We re-organized this sentence in the revised manuscript as follows "The paddy plots received water through**

precipitation and stream water irrigation, while evaporation to air and seepage to the subsoil were the primary vectors for water loss. There was no direct runoff from either paddy." (page 5 lines 16-18)

(11) this is not the minus sign. Consider replacing all - by –

**We revised the manuscript very carefully, and all the "-" was changed to "-" throughout the manuscript.**

(12) can you clarify; do you mean the variability between the triplicate samples?

**Yes, we re-worked this sentence in the revised manuscript as follows: "The variability between the triplicate samples were less than 7.5% for THg and MeHg analysis for both water and soil samples." (page 9 lines 14-15)**

(13) I don't understand this sentence. Hg in precipitation is equal or closely linked to "wet deposition", while dry deposition is another process.

**As shown in the manuscript both dry and wet atmospheric deposition were collected concurrently with the TGM measurement once every 20 days using this sampling method (see page 6 lines 11-13). Therefore, we defined the precipitation as "wet and dry deposition" in the revised manuscript (see page 10 line 9).**

(14) what about the ratio for the Huaxi regional background, which is quite high for paddy water ? Any explanation ?

**In comparison, the $MeHg_{unf}/HgT_{unf}$ ratios in precipitation (0.76±0.41 %), irrigation water (2.2±0.98 %), and paddy water (10±7.9 %) were relatively elevated, probably due to the lower $HgT_{unf}$ concentrations in the corresponding samples (Meng et al., 2011). We added these sentences in the revised manuscript (see page 11 lines 13-16)**

(15) was the difference significant ? K-W test ?

**Yes, significant difference was observed ($p<0.01$). The detailed information concerning**

statistical analysis data was added in the revised manuscript as follows: "K-W test, p<0.01" (see page 12 line 2).

(16) the term "mechanistic relationship" is too vague. If methylation is active, is it expected that $HgT_f$ and $MeHg_f$ are correlated? (was it the case in your previous studies ? Meng et al 2014+ check literature). Then, if methylation is an important process at the artisanal mining site only, you can state it and try to explain why it is, and why not at the other site.

We believe that net MeHg production is principally governed by the supply of fresh deposited Hg to soil (page 19 lines 10-11). Restricted supply of newly deposited Hg to depths below the soil-water interface is a plausible explanation for the sharply reducing concentration of MeHg with depth at Gouxi; newly deposited Hg is constrained to surface soil and cannot be transferred to lower depth. Therefore, a direct positive relationship between $HgT_f$ and $MeHg_f$ concentrations in soil pore water was observed at Gouxi during the rice growing season (page 18 lines 6-10).

The Wukeng site has received significant historic Hg deposition as a function of large scale mining, but is not currently receiving significant inputs of fresh Hg. Atmosphere-derived mercury is physically unstable and bioavailable when it first enters the rice paddy (Hintelmann et al., 2002; Schuster, 2011). Immediate reactions of this new Hg with soil constituents are governed by adsorption-desorption interactions with soil surfaces (Schuster, 1991) which favour the retention of Hg in the surface layers of the soil profile. Over time this newly deposited Hg will be transformed into more stable, less available forms (Schuster, 1991), and the net methylation potential of this Hg will consequently decrease. The relatively low MeHg concentration in soil at Wukeng is indicative of old Hg which has become tightly bound to soil complexes over time, and is unavailable for methylation (Hintelmann et al., 2002). Consequently, there is no correlation between $HgT_f$ and $MeHg_f$ in soil pore water at Wukeng (see section 3.3.1). (see page 18 lines 14-24).

(17) I also believe this, but please insert references supporting this, as observed in other paddy field studies. Also, although it is implicitly stated, complete the sentence by reminding that after Day 80 the field is no more flooded, and hence methylation is probably stopped. --

edit OK I see now it is discussed a little bit further. Then, this sentence should be (re)moved, so that all interpretation is put together (not a bit here, and the rest further in the text)

**The authors definitely agree with the reviewer's comments herein. We deleted this sentence in the revised manuscript. More detailed explanation was added in the revised manuscript as follows:**

**Changing redox parameters over the rice growing season may affect the process of Hg methylation. Previous studies have observed that in artificially Hg-polluted soil, Hg bioavailability to methylation can be significantly affected by the level of water saturation (Rothenberg and Feng, 2012; Wang et al., 2014; Peng et al., 2012). Peng et al. (2012) specified that intermittent flooding, as opposed to continuous flooding, could reduce soluble Hg concentrations and inhibit Hg methylation in the rice rhizosphere, subsequently decreasing the accumulation of MeHg in rice grain. Flooded conditions enhance anaerobic microbial activities and increase MeHg yields. Drying of a paddy field is an important cultivation step to control rice plant tillering and increase yield. Therefore, one possible reason for the considerably elevated MeHg concentrations in soil at Gouxi between Day 20 and Day 80 relative to Day 100 is an enhancement of Hg bioavailability and numbers of SRB under flooded conditions that stimulated Hg methylation, and increased the soil MeHg concentration (Wang et al., 2014). As the paddy field dried from Day 80, some degree of net MeHg degradation may have occurred, which could be attributed to the decreased SRB numbers and proportion of Hg methylators in the rhizosphere under aerobic conditions(Wang et al., 2014). This likely contributed to a decreasing trend in soil MeHg concentration during the harvest period. (see page 14 lines 9-23).**

(18) be more specific about the processes. E.g. redox conditions change when the file dis dried, therefore MeHg degradation occurs. + give references of the biochemical processes taking place.

**Yes, we re-organized this sentence in the revised manuscript as follows: As the paddy field dried from Day 80, some degree of net MeHg degradation may have occurred, which could be attributed to the decreased SRB numbers and proportion of Hg methylators in the rhizosphere under aerobic conditions(Wang et al., 2014) (see page 14 lines 20-22)**

(19) these references are relevant to a certain extent, but treated forest soils only. Are there references specific to paddy fields for this aspect ?

Yes, the authors agree with the reviewer's comments herein. We re-worked the model in the revised manuscript (see section 3.4 and response to comment #1). The references mentioned in this comment were removed from manuscript.

(20) not sure that "native" is the appropriate word. Native makes me think to geogenic i.e. "natural" Hg, while here it is mostly from anthropogenic sources. Consider rephrasing.

Yes, the "native Hg" was change to "old Hg" in the revised manuscript. Furthermore, "old Hg" was defined as "Mercury already present in the soil is termed 'old Hg', which can be either of geogenic and anthropogenic origin" in the revised manuscript (page 15 line 7-9).

(21) is this water accumulated in the rice ??? then 34 cm seems a lot ! but unfortunately I cannot check Lan et 2010. The amount of transpiration seems very, very low if I compare e.g. to Brunel et al (1992) WRR 28 (5):1407-1416.

Very sorry for our stupid mistake. After re-checked this data from the cited literature, we found that $M_d$ is very low and needn't to be considered in this model. Hence, we removed $M_d$ from equation 4. Furthermore, we re-calculated the data in the revised manuscript (page 16 lines 5-11).

(22) which? apart from runoff, what else could it be ? draining the paddy field ?

Mo is the cumulative amount of water lost by other pathways (e.g. animal activities and draining the paddy during the ripening period). We reworked this sentence in the revised manuscript. (see page 16 lines 8-9)

(23) MAJOR COMMENT.
You need to take into account that the pool of 'old' Hg is probably constituted (partly) by deposition and irrigation from previous years. The Hg balance that you implemented compares 1 year of Hg input via deposition and irrigation to Hg accumulation over XX years. Is this the purpose of the Hg balance model? What kind of useful information does this bring?

Yes, the authors definitely agree with the reviewer's comments herein. After carefully consideration, we re-worked the model in the revised manuscript (see detail in section 3.4 and the response to comment #1).

(25) what if the Wukeng soil had had a low pH more favorable to methylation ? The important difference of pH between the two sites prevents in my opinion any conclusion regarding a possible difference between old / fresh Hg available for methylation. If the redox and pH conditions are not good for methylation, it will not occur (whatever the 'type' of Hg present in the soil). I recommend to discuss this, and reformulate the conclusions taking this into account.

**This comment is same to comment #2. Please go back to the detailed response to comment #2.**

(26) this Table contains a lot of information but is difficult to read. One cannot easily see to which sample matrix each result belongs (I think the text in left column should be vertically aligned to the top - this might already improve the readability but please try to improve this Table).

**Yes, we re-worked Table 2 based on the reviewer's comments in the revised manuscript (see page 28 lines 1-7 and page 29 lines 1-6)**

(27) increase slightly axis tick label and legend font size.

**Yes, we re-worked Figure 2 according the reviewer's comments in the revised manuscript (page 33 lines 1-4)**

---

## Author Comment (AC4) · 25 Mar 2016

The comment was uploaded in the form of a supplement:
http://www.biogeosciences-discuss.net/bg-2015-638/bg-2015-638-AC4-
supplement.pdf

---

## Author Response (AR1)

**Response to comments from reviewers 1**

The study conducted by Zhao et al. takes on a challenging set of important questions regarding mercury in rice paddy soils. The study looks at two different systems, a point source polluted rice paddy at Gouxi, where artisanal mining of Hg occurs, and Wukeng, an abandoned mine with reduced atmospheric deposition of Hg. These two sites are commonly compared to Hauxi, a 'control site' of regional pollution. The design of the study is less-than ideal (see below in specific comments) but the temporal measurements of MeHg makes this study an asset for those interested in Hg in rice paddies.

**On behalf of my co-authors, I would like to thank the anonymous reviewer for dedicating time to provide comments and criticism. The reviewer raises important issues, which have helped us to improve the manuscript. We have completed a substantial revision and our changes are highlighted in red color in the revised manuscript. Our point-by-point response to the reviewer's comments is as follows:**

The manuscript falls short in a few areas. First, the paper lacks a strong, clear statement of hypotheses and the biological/chemical mechanism. By clearly stating the hypotheses, this will greatly aid in organizing the discussion and determine which mechanisms need to be specifically addressed in greater detail in the discussion. From my understanding of the manuscript one hypothesis should be "We hypothesize that theGouxi site will have greater MeHg than the Wukeng site because greater atmospheric deposition of 'new Hg' is more susceptible to methylation". A second hypothesis could be "We expect greater MeHg in the upper mineral soil horizons at the Gouxi site than Wukeng site because 'old Hg' is less susceptible to methylation". These are just suggestions but explicitly stating them is needed to guide the discussion.

**We agree with the reviewer's comment that the current manuscript needs a strong and clear-statement hypothesis. On the basis of the reviewer's suggestion we have added hypotheses in the revised manuscript as follows: "The hypothesis of our study, established from existing literature and data we have collected, is that the MeHg concentration in Gouxi soil will be greater than Wukeng due to the a higher flux of IHg in the atmosphere at Gouxi through current-day Hg smelting. Furthermore, we believe that the MeHg concentration will be greater in surface soil, as this is the receiving environment for freshly deposited IHg" (see page 14 line 27 and page 15 lines 1-4). For emphasis we have added these new statements to the Discussion section and believe that these follow on from the clear aim of the work stated in the Introduction.**

In addition to the missing hypotheses, the inclusion of the water-atmospheric model was unnecessary in lieu of a more simplistic comparison of fluxes. The modeling made many assumptions that also made it unreasonable to apply (see specific comments).

**Thank you for this suggestion. We have provided a detailed response to this comment in our following**

responses to the specific comments raised in this review.

Lastly, Hg in Oryza sativa data should be presented in this study. Since the uptake of MeHg and Hg byrice is central to this study, linking the belowground processes to the plant would be agreat addition to the study.

**The reviewer makes a valid point, however sampling of plant samples was not part of the design of this particular work (a PhD study at the Institute of Geochemistry of the Chinese Academy of Sciences). The current study focused on the speciation and distribution of Hg in the paddy soil during the rice growing season based on the previous published finding that the MeHg concentration in rice was significantly different between the two study areas. Rice plant samples were therefore not collected as part of this study (page 5 lines 20-22) which is focused solely on the soil system. Meng et al. (2010) focused on the Wanshan area of China, a region of both historical large-scale and current small-scale mercury mining and showed that the MeHg concentration in rice grain collected from an active artisanal Hg mining areas (32±14 ng g$^{-1}$) was significantly higher than in rice grain collected from an abandoned Hg mining area (7.0±3.2 ng g$^{-1}$). These sentences shown above were added in the revised manuscript (see page 3 lines 11-15). It was this earlier report of MeHg in rice the provided the justification for the soil study we report in this paper. We have taken this point under consideration and will look to resample and review the MeHg concentration in rice in the two areas, and correlate this with soil properties in a future study.**

**Specific comments**

The introduction lacks discussion of microbial and chemical mechanisms responsible for methylation. Even if you are not testing for methylating bacteria, the mechanisms of methylation should not be absent.

**We agree with the reviewer's comments here. An overview of the microbial and chemical mechanisms responsible for methylation was added to the revised manuscript as follows:**

**Current understanding is that the mobility and methylation of Hg in ephemeral flooded soil is determined by a range of factors, such as redox potential, pH, dissolved organic carbon, sulfur, iron, and dissolved Hg content (e.g., Ullrich et al., 2001; Benoit et al., 2001). Mercury methylation is largely facilitated by a subset of sulfate-reducing bacteria (SRB) (Gilmour et al., 1992) and/or iron-reducing bacteria (Fleming et al., 2006) in anoxic conditions. Specially, the methylation of inorganic Hg (IHg) in paddy soil primarily occurs through a process mediated by sulfate-reducing bacteria (Peng et al., 2012; Rothenberg and Feng, 2012; Wang et al., 2014; Liu et al., 2014; Liu et al., 2009; Somenahally et al., 2011) (see page 2 lines 22-27 and page 3 lines 1-2).**

In the experimental design, I understand the practicality of monitoring two, 10 X 10 meter rice paddies. However, sampling one plot multiple times to assess an treatment/affect is potentially pseudoreplication.

We have considered the point of the reviewer. As summarized in the manuscript "Changing redox parameters over the rice growing season may affect the process of Hg methylation. Previous studies have observed that in artificially Hg-polluted soil, Hg bioavailability for methylation can be significantly affected by the level of water saturation (Rothenberg and Feng, 2012; Wang et al., 2014; Peng et al., 2012). Peng et al. (2012) specified that intermittent flooding, as opposed to continuous flooding, could reduce soluble Hg concentrations and inhibit Hg methylation in the rice rhizosphere, subsequently decreasing the accumulation of MeHg in rice grain. Flooded conditions enhance anaerobic microbial activities and increase MeHg yields. The drying of a paddy field is an important cultivation step to control rice plant tillering and increase yield." (page 14 lines 1-8).

In this study the two selected experimental rice paddy were cultivated during the period 1st June through 10th September (100 days) 2012. Standing water (2-8 cm) was maintained above the soil surface (flooded condition) throughout the growing period, from Day 0 to Day 80. The paddy fields were thereafter drained from Day 80, prior to harvest between Days 90 to 100. During the 10 day draining period, approximately 2-4 cm depth of water was maintained above the soil surface (page 5 lines 6-10). In order to investigate the speciation, distribution, and Hg methylation in the paddy soil during the whole rice growing season, five consecutive sampling campaigns were conducted during the rice growing season (1st June-10th September, 2012) (page 5 lines 14-15). Given the experimental design and the aims of this study, we are of the opinion that sampling one plot multiple times is necessary, and is not pseudoreplication. We do hope that the reviewer will review his comments here.

For Section 3.3.2 on Soil Cores, the physical and chemical data of the soil cores should be included in the discussion.

The authors agree with the reviewer's comments here. Considerably more detail on the physical and chemical data of the soil cores is available, but we believe that presenting this data is beyond the scope of the current manuscript and would un-necessarily add length to the current work. Instead we have chosen to present the soil physio-chemical data in a stand-alone paper. To add to the current manuscript, the key data from this 'companion paper' is summarized in the revised manuscript. Specifically:

Methylation can be affected by the pH and organic matter content of soil, and an analysis of soil physiochemical parameters in the soil cores of this study is reported in a companion paper (Zhao et al., 2016). Briefly, the mean organic matter in soil cores was 4.8±0.75 % and 3.5±0.59 % at Gouxi and Wukeng, respectively. The mean soil pH was the same for both sites (6.7±0.10 at Gouxi and 6.6±0.14 at Wukeng) and did not change as a function of sampling time, despite the variation reported for irrigation water and paddy water at Wukeng in the current study (Table 5). The consistency of soil pH throughout the sampling period indicates that irrigation water and paddy water have little influence on bulk soil pH. Statistical analysis revealed that there is no direct impact of pH and organic matter content on the MeHg concentration in soil across the two sampling sites, indicating that absolute pH and organic matter might not be the most important factors regulating Hg methylation activity (Zhao et al., 2016) (see page 13 lines 13-28).

Was dissolved oxygen, sulfate, Fe or other important electron acceptors measured and comparable through time?

Yes. As described in the previous response point, this data is available and has been both presented in a companion paper, and is now summarized in the current paper:

In order to better understand the factors controlling Hg methylation in rice paddy soil, the concentration of $Fe^{2+}$, $Fe^{3+}$, $S^{2-}$, and $SO_4^{2-}$ in soil pore water was determined and this data is described, in detail, in a companion paper (Zhao et al., 2016). Briefly, no discernible vertical trend in $Fe^{3+}$ distribution was observed in the soil pore water across the two sampling sites during the sampling period. The $Fe^{2+}$ concentrations in soil pore water at Gouxi exhibited a narrow range (41~417 µM), relative to that at Wukeng (2.3~843 µM). The $S^{2-}$ concentration in the soil pore water showed limited variation with depth at Wukeng (mean=0.70±0.36 µM, range=0.07~1.2 µM) relative to Gouxi (mean=1.8±0.79 µM, range=0.69~3.8 µM), with the highest value recorded in the surface soil layer at both sites. Temporal variation of sulfide concentrations at Wukeng and Gouxi was insignificant (K-W test, p=0.73 and p=0.33 for Wukeng and Gouxi, respectively). The highest $SO_4^{2-}$ concentrations were recorded in the surface soil layer and decreased with depth across the two sampling sites. As described in the companion paper (Zhao et al., 2016), $SO_4^{2-}$ stimulation of SRB activity was a potentially important metabolic pathway for Hg methylation in the rice paddy soil at the two Hg mining sites, while iron cycling in the rice paddies could impact the availability of Hg in pore water for methylation. This information was added in the revised manuscript (page 12 lines 10-23).

Page 13 Line 1 – 9: The results should be better integrated with existing knowledge about the effect of microbial production of MeHg in flooded soils.

The authors agree with the reviewer's comments here. Therefore, we re-organized this paragraph as follows:

Changing redox parameters over the rice growing season may affect the process of Hg methylation. Previous studies have observed that in artificially Hg-polluted soil, Hg bioavailability for methylation can be significantly affected by the level of water saturation (Rothenberg and Feng, 2012; Wang et al., 2014; Peng et al., 2012). Peng et al. (2012) specified that intermittent flooding, as opposed to continuous flooding, could reduce soluble Hg concentrations and inhibit Hg methylation in the rice rhizosphere, subsequently decreasing the accumulation of MeHg in rice grain. Flooded conditions enhance anaerobic microbial activities and increase MeHg yields. The drying of a paddy field is an important cultivation step to control rice plant tillering and increase yield. Therefore, one possible reason for the considerably elevated MeHg concentrations in soil at Gouxi between Day 20 and Day 80 relative to Day 100 is an enhancement of Hg bioavailability and numbers of SRB under flooded conditions that stimulated Hg methylation, and increased the soil MeHg concentration (Wang et al., 2014). As the paddy field dried from Day 80, some degree of net MeHg degradation may have occurred, which could be attributed to the decreased SRB numbers and proportion of Hg methylators in the rhizosphere under aerobic conditions (Wang et al., 2014). This could have contributed to a decreasing trend in soil MeHg concentration during the harvest period (see page 14 lines 1-15)

The model is an interesting thought-experiment based on a number of assumptions such as negligible amounts of Hg volatilizing from the water surface and dynamic equilibrium of the aqueous solution. However, the assumption I find the hardest to justify is that the system is behaving as an unsaturated soil (as cited in Munthe and Hintelmann) and not behaving as a water-sediment system. I understand it simplifies the system to a traditional unsaturated agricultural system but the fact is the water and saturated soil (now behaving as a sediment) are exchanging with each other rather than acting as one system. Instead of the model in 3.4, is it possible to just compare the atmospheric Hg fluxes, irrigation Hg fluxes, and 'Old Hg' pools and find the same conclusion that Hg was primarily from atmospheric deposition and MeHg is produced in situ?

We fully agree with the reviewer's comments here. After careful consideration of the revised manuscript and the reviewer's suggestion, we have simplified the model in the revised manuscript (see detail in section 3.4 page 14 lines 24-27 and page 15 lines 1-26, page 16 lines 1-26).

Briefly, the relative flux of the different sources of Hg (THg and MeHg) to the rice paddy soil during the rice growing season was calculated. Furthermore, the amount of native THg and MeHg present in the paddy soil (20 cm depth) was calculated. The calculated data are listed in Table 4. The calculations showed that the MeHg flux to the rice paddy soil attributable to atmospheric deposition (Gouxi=3.3 mg ha$^{-1}$; Wukeng=2.1 mg ha$^{-1}$) and irrigation (Gouxi=1.8 mg ha$^{-1}$; Wukeng=4.2 mg ha$^{-1}$) was 3 orders of magnitude smaller than the amount of native MeHg already present in the paddy soil (Gouxi=2026 mg ha$^{-1}$, Wukeng=1613 mg ha$^{-1}$). A similarly low value for atmospheric deposition (Gouxi=$1.8\times10^{-2}$ mg ha$^{-1}$; Wukeng=$3.1\times10^{-3}$ mg ha$^{-1}$) and irrigation water (Gouxi=$0.39\times10^{-3}$ mg ha$^{-1}$; Wukeng=$1.3\times10^{-3}$ mg ha$^{-1}$) flux was apparent for THg (Table 4) when compared with the soil THg pool (Gouxi=3.2 mg ha$^{-1}$; Wukeng=32 mg ha$^{-1}$). Our calculations therefore suggest that despite the highly elevated THg concentration in atmospheric deposition and irrigation water, the flux of new Hg (MeHg and THg) from external sources was small because of the relatively large pool of old Hg in soil (Dai et al., 2013). Therefore, we propose that the dominant source of MeHg to the paddy soil is in situ methylation of inorganic Hg.

Statistical analysis showed that the THg flux from atmospheric deposition was significantly higher than from irrigation across the two sampling sites (K-S test, $p<0.001$). Furthermore, the THg atmospheric deposition flux at Gouxi was approximately 6 times higher than at Wukeng during the rice growing season. Therefore, we propose that the flux of THg to paddy soil at Gouxi was primarily due to atmospheric deposition associated with ongoing artisanal Hg activities, in agreement with the hypothesis of our study. (see page 16 lines 7-26 and page 17 lines 1-3)

Page 15 Line 25: Could you compare your data on estimated Hg methylation with other rice paddies in Asia to assess how alkaline conditions have slowed or retarded Hg methylation?

This is a very good question. While we are unable to cite data for other paddies in Asia we have re-organized this paragraph to include the following information in the revised manuscript:

The mean concentration of $HgT_f$ in paddy water at Wukeng ($197\pm78$ ng $L^{-1}$) was proximately 2 times higher than that at Gouxi ($105\pm58$ ng $L^{-1}$), whereas the $MeHg_f$ concentration in paddy water at Gouxi ($4.7\pm4.2$ ng $L^{-1}$) was approximately 8 times higher than that at Wukeng ($0.62\pm0.29$ ng $L^{-1}$) (Table 3). Furthermore, the concentration of $MeHg_f$ in paddy water at Wukeng ($0.62\pm0.29$ ng $L^{-1}$) was significantly higher than that in precipitation ($0.14\pm0.07$ ng $L^{-1}$), but significantly lower than in irrigation water ($0.96\pm0.50$ ng $L^{-1}$) and soil pore water ($1.7\pm0.88$ ng $L^{-1}$) in the soil surface layer during the rice growing season (K-S test, $p<0.001$) (Table 2 and Table 3). Generally, there are three possible sources of MeHg in the paddy water: 1) in situ production being controlled by chemistry condition (e.g. redox and pH), 2) diffusion of MeHg from underlying soil, and 3) MeHg flux of atmospheric deposition and irrigation. We propose that baseline $MeHg_f$ in paddy water at Wukeng is primarily due to the diffusion of MeHg from the surface layer of sediment and MeHg flux from atmospheric deposition and irrigation.

The sampling site for the Wukeng paddy was located next to a calcine pile and the proximity of this waste had a major impact on water chemistry. Both the irrigation water (pH=$11\pm0.45$) and paddy water (pH=$8.6\pm1.3$) were alkaline during the rice growing season (Table 5). We suggest that the alkaline conditions of the irrigation at Wukeng could restrain Hg methylation and/or stimulate MeHg demethylation in paddy water (Ullrich et al., 2001). Rothenberg et al. (2012) reported that alkaline paddy water (pH >11) at highly-contaminated mining sites can restrain the bioavailability of $Hg^{2+}$ for Hg methylation, resulting in lower pore water and soil MeHg concentrations despite higher total Hg concentrations. The findings of our study are in agreement with those of Rothenberg et al. (2012).

In contrast, the $MeHg_f$ concentration in paddy water at Gouxi ($4.7\pm4.2$ ng $L^{-1}$) was significantly higher than in precipitation ($0.33\pm0.17$ ng $L^{-1}$) and irrigation water ($0.31\pm0.30$ ng $L^{-1}$), but significantly lower than in soil pore water ($7.8\pm5.2$ ng $L^{-1}$) in the soil surface layer during the rice growing season (K-S test, $p<0.001$) (Table 2 and Table

3), with the data at Day 80 as an exception. The maximum MeHg$_f$ concentration was not recorded for the surface soil pore water (3.6 ng L$^{-1}$) but for the paddy water (4.7 ng L$^{-1}$) at Day 80. The implication is that MeHg in this region is not only due to MeHg diffusion from surface soil and/or the MeHg flux of atmospheric precipitation and irrigation, but also from in situ methylation in anoxic water with relatively low pH (pH=6.9 on Day 80) (Table 5). Gilmour and Henry (1991) specified that low pH and anaerobic condition not only increase methylation rates but also decrease demethylation rates, resulting in net production of MeHg. Both paddy water and irrigation water at Gouxi exhibit pH and redox conditions that can be considered optimal for Hg methylation (Table 5), favouring net methylation in the paddy water (Ullrich et al., 2001). Active Hg methylation within the Gouxi rice paddy is implied in this study. However, data, to directly support this hypothesis are limited. To better understand this observation, further work needs to be done  (page 17 lines 2-28 and page 18 lines 1-7)

**Technical comments**

Page 2 Line 10: a strong bioaccumulator? There is an adjective missing. I might suggest re-writing the sentence.

Yes, we re-worked this sentence in the revised manuscript as follows: Reports of methyl mercury (MeHg) contamination of rice grain (*Oryza sativa*) have recently focussed scientific attention on this important agricultural crop (see page 2 lines 11-12).

Page 2 Line 15: In which part does the rice uptake Hg?

We re-organized this paragraph in the revised manuscript as follows: Numerous studies have reported high MeHg concentrations in rice grain collected from Indonesia (Krisnayanti et al., 2012) and different parts of China (Horvat et al., 2003; Qiu et al., 2008; Meng et al., 2014). The MeHg concentration in rice grain (brown rice) can be enhanced even in cases where soil is not significantly elevated in Hg (Zhang et al., 2010a; Horvat et al., 2003). Meng et al. (2014) specified that the majority (~80 %) of MeHg was found in edible white rice (see page 2 lines 13-17)

Page 2 Line 27: Although its a common term in Hg literature, please define IHg.

Yes, we defined IHg as "inorganic Hg" in the revised manuscript. (see page 2 line 27)

Page 3 Line 21-25: I feel these details should be in the methods since they describe your actions.

Yes, these sentences were moved to section 2.2 (sample collection and preparation) in the revised manuscript.

Page 4: Is it possible to provide coarse latitude and longitude for the Wanshan mining district in the text?

**Yes, we added latitude and longitude of the Wanshan mining district in the revised manuscript (see page 4 line 7).**

Page 7 Line 3: The phrase 'under argon' is in exact. Please re-phrase with details.

**Yes, detailed information was added in the revised manuscript as follows: Firstly, the air (oxygen) in the glove bag was eliminated manually. Then, the pure argon from a portable argon tank was injected into the glove bag through a Teflon tubing. This information was added in the revised manuscript. (see page 7 lines 11-12)**

Page 7 Line 10: Extra period at beginning of sentence.

**Yes, we re-worded this sentence in the revised manuscript as follows "At each sampling time (Days 0, 20, 40, 60, and 80) a second soil core was collected and immediately placed into liquid nitrogen" (see page 7 lines 19-20)**

Page 8 Line 10: correct to EPA method 1630.

**Yes, we re-worked this sentence in the revised manuscript as follows: "following EPA method 1630 (U.S. EPA, 2001)" (see page 8 line 15-16)**

Page 8 Line 12: Please define $HgT_{unf}$ and $HgT_f$ explicitly at first use.

**Yes we defined $HgT_{unf}$ and $HgT_f$ as total Hg ($HgT_{unf}$) and dissolved total Hg ($HgT_f$), respectively, in the revised manuscript (see page 6 line 9 and page 6 line 11-12).**

Page 8 Line 26-Page9 Line 1: Please include "respectively" to indicate the relationship between the blank concentrations with THg and MeHg.

**Yes, we added "respectively" in the corresponding sentence in the revised manuscript (see page 9 line 6).**

Page 9 Line 13-15: Please mention these are non-parametric tests for those unfamiliar with those tests.

**Yes, we added the information "non-parametric tests" in the revised manuscript. (see page 9 lines 19-21)**

Pages 9-15: It is conventional for this journal to include a space between numbers and symbols, particularly when expressing the mean and standard deviation.

**Yes, we revised the manuscript very carefully and added "space" between numbers and symbols throughout the manuscript.**

Page 15 Line 6: Was this model used for Hg and MeHg?

**This model was used for THg and MeHg in the soil cores. We re-worked this sentence in the revised manuscript as follows "Using Equations 1-2, the relative flux of the different sources of Hg (THg and MeHg) to the rice paddy soil during the rice growing season was calculated" (page 16 line 7-8)**

**Response to comments from reviewers 2**

**GENERAL COMMENTS**

The study by Zhao et al. addresses Hg contamination and mthylation in paddy fields in the province of Guizhou, China. This is a topic of high importance for human and ecosystem health in paddy field areas. The study is relevant for Biogeosciences and falls within the Aims and scope. The methods are well explained. The results are well presented, although the clarity of the Tables should be improved. The discussion is generally good, but could benefit from more links to existing literature. I have two major comments that should be addressed to improve the paper:

**On behalf of my co-authors I sincerely thank the anonymous reviewer for dedicating their time to provide comments and criticism. The reviewer raises many important issues. My co-authors and I have considered these and made appropriate changes to the text, and we are confident the manuscript has been significantly improved as a result. Revisions are shown in red color in the revised manuscript, with our point-by-point response presented as follows.**

(1) the Hg balance model (eq. 3 and 4). I don't really see the added value of this model. Moreover, some assumptions are very strong (eg rice transpiration amount extremely low), and the athors compare the input of 'fresh' Hg (irrigation and deposition) of 1 year, to the 'old' Hg pool accumulated over the years. Therefore the Hg balance results entirely depend on the number of years during which Hg has accumulated in the surface layer (X years of paddy field irrigation, etc.). I recommend to completely revise the model or simply drop it (unless you can clearly demonstrate what it brings to the discussion and how it supports your conclusions)

**We agree with the reviewer's comments here. After carefully considering Reviewer 2's comments in combination with those of Reviewer 1, we have simplified the model in the revised manuscript (see detail in section 3.4 page 14 lines 24-27 and page 15 lines 1-26, page 16 lines 1-26).**

**Briefly, the relative flux of the different sources of Hg (THg and MeHg) to the rice paddy soil during the rice growing season was calculated. Furthermore, the amount of native THg and MeHg present in the paddy soil (20 cm depth) was calculated. The calculated data are listed in Table 4. The calculations showed that the MeHg flux to the rice paddy soil attributable to atmospheric deposition (Gouxi=3.3 mg ha$^{-1}$; Wukeng=2.1 mg ha$^{-1}$) and irrigation (Gouxi=1.8 mg ha$^{-1}$; Wukeng=4.2 mg ha$^{-1}$) was 3 orders of magnitude smaller than the amount of native MeHg already present in the paddy soil (Gouxi=2026 mg ha$^{-1}$, Wukeng=1613 mg ha$^{-1}$). A similarly low value for atmospheric deposition (Gouxi=1.8×10$^{-2}$ mg ha$^{-1}$; Wukeng=3.1×10$^{-3}$ mg ha$^{-1}$) and irrigation water (Gouxi=0.39×10$^{-3}$ mg ha$^{-1}$; Wukeng=1.3×10$^{-3}$ mg ha$^{-1}$) flux was apparent for THg (Table 4) when compared with the soil THg pool (Gouxi=3.2 mg ha$^{-1}$; Wukeng=32 mg ha$^{-1}$). Our calculations therefore suggest that despite the highly elevated THg concentration in atmospheric deposition and irrigation water, the flux of new Hg (MeHg and THg) from external**

sources was small because of the relatively large pool of old Hg in soil (Dai et al., 2013). Therefore, we propose that the dominant source of MeHg to the paddy soil is in situ methylation of inorganic Hg.

Statistical analysis showed that the THg flux from atmospheric deposition was significantly higher than from irrigation across the two sampling sites (K-S test, p<0.001). Furthermore, the THg atmospheric deposition flux at Gouxi was approximately 6 times higher than at Wukeng during the rice growing season. Therefore, we propose that the flux of THg to paddy soil at Gouxi was primarily due to atmospheric deposition associated with ongoing artisanal Hg activities, in agreement with the hypothesis of our study. (see page 16 lines 7-26 and page 17 lines 1-3).

(2) the authors should pay more attention and discuss in more details to the biochemical processes affecting Hg methylation. What if the Wukeng soil had had a low pH more favorable to methylation? Would the conclusions of historical vs. artisanal Hg mining still hold? The important pH difference between the two sites prevents any conclusion regarding the impact on methylation of the "type" of Hg available (old at Wukeng vs fresh at Gouxi). If the redox and pH conditions are not good for methylation, it will not occur (whatever the 'type' of Hg present in the soil). I strongly recommend to discuss this (with additional literature references), and reformulate the conclusions taking this into account.

We agree with the reviewer's comments here. Based on the reviewer's comments we have reorganized the discussion in section 3.3, section 3.5 and section 3.6 in the revised manuscript. Note that we cite a companion paper which is currently under review in Environmental Pollution. The data set was simply too large and complex for a single paper and we have split our findings into what we believe are two discrete and scientifically sound manuscripts. In response to the reviewers' concerns we briefly summarise the key data from the companion paper that is relevant to the current paper in the current paper:

1) Section 3.3.2 (page 13 lines 18-28, page 14 lines 1-14).

[revised manuscript text omitted]

**SPECIFIC COMMENTS and TECHNICAL COMMENTS**

(1) mention somewhere in the abstract the location (China, and at least the province)

**Yes, the detailed location "Guizhou province China" was added in the revised manuscript (see page 1 lines 18)**

(2) use "inorganic Hg" (or define the abbreviation earlier)

**Yes, we defined IHg as inorganic Hg in the revised manuscript. (page2 line 1)**

(3) sentence is unclear "in situ production of MeHg is dependent on elevated IHg in the atmosphere and on the deposition of"... ?

**Yes, we re-organized this sentence in the revised manuscript as "We propose that the in situ production of**

MeHg in paddy water and surface soil is dependent on elevated Hg in the atmosphere and the consequential deposition of new Hg into a low pH anoxic geochemical system" (see page 2 lines 3-5).

(4) put the references chronologically.

**Yes we re-organized references as (Horvat et al., 2003; Qiu et al., 2008; Meng et al., 2014) in the revised manuscript (page 2 lines 14-15).**

(5) to support the assumed "consensus", more than one author should be provided.

**Yes, two more references (Qiu et al., 2012; Yin et al., 2013) were added in the revised manuscript (page 3 line 9)**

(6) large scale / small-scale: be consistent with the "-"

**Yes, "large scale" was changed to "large-scale" in the revised manuscript (see page 3 line 12).**

(7) these are all the possible compartments where Hg can be found... this sounds like an "empty" argument or too generic

**We agree with the reviewer's comments here. Therefore, we have reorganized this paragraph in the revised manuscript as follows: "Meng et al. (2010) focused on the Wanshan area of China, a region of both historical large-scale and current small-scale mercury mining and showed that the MeHg concentration in rice grain collected from an active artisanal Hg mining areas ($32\pm14$ ng g$^{-1}$) was significantly higher than in rice grain collected from an abandoned Hg mining area ($7.0\pm3.2$ ng g$^{-1}$). Such studies on MeHg and rice emphasize that factors which control the biochemical cycling of Hg within rice paddy ecosystems are very complex, and include the concentration and distribution of Hg in ambient air, wet/dry deposition, irrigation water, and the solid and liquid phases of soil" (see page 3 lines 11-18)**

(8) confusing. What is "assessing the status of Hg species" ?consider rephrasing ("we analyzed Hg speciation in ...")

**We agree with the reviewer's comments here. We re-worked this sentence in the revised manuscript as follows: "The biochemical processes that control the cycling of Hg in paddy soils impacted by Hg mining are poorly understood. The objectives of the current study were therefore to 1) investigate the speciation and distribution of**

**Hg in paddy soil, and 2) assess the primary source and mechanism for Hg methylation within a Hg mining area."** (see page 3 lines 23-26).

(9) if I'm correct you investigate methylation, not other transformations. Then replace by "Hg methylation"

**Yes, we re-worked this sentence in the revised manuscript as follows "assess the primary source and mechanism for Hg methylation within a Hg mining area" (see page 3 lines 26)**

(10) can you be more precise about what you consider to be seepage and outflow? Is seepage infiltration to the subsoil ? Does outflow mean runoff ?

**Yes, seepage infiltration is to the subsoil and outflow means runoff. We re-organized this sentence in the revised manuscript as follows "The paddy plots received water through precipitation and stream water irrigation, while evaporation to air and seepage to the subsoil were the primary vectors for water loss. There was no direct runoff from either paddy" (page 5 lines 10-12)**

(11) this is not the minus sign. Consider replacing all - by –

**Thank you for pointing this out. We revised the manuscript very carefully, and all the "-" was changed to "-" throughout the manuscript.**

(12) can you clarify; do you mean the variability between the triplicate samples?

**Yes, we re-worked this sentence in the revised manuscript as follows: "The variability between the triplicate samples was less than 7.5 % for THg and MeHg analysis for both water and soil samples" (page 9 lines 8-9)**

(13) I don't understand this sentence. Hg in precipitation is equal or closely linked to "wet deposition", while dry deposition is another process.

**As shown in the manuscript "both dry and wet atmospheric deposition were collected concurrently with the TGM measurement once every 20 days using this sampling method" (see page 6 lines 6-8). Therefore, we again defined the precipitation as "wet and dry deposition" in the revised manuscript (see page 10 line 4).**

(14) what about the ratio for the Huaxi regional background, which is quite high for paddy water ? Any explanation?

The MeHg$_{unf}$/HgT$_{unf}$ ratios for precipitation (0.76±0.41 %), irrigation water (2.2±0.98 %), and paddy water (10±7.9 %) for both mining sites were elevated relative to the regional background, and we believe this is due to the lower HgT$_{unf}$ concentration reported for the regional background. We added these sentences in the revised manuscript (see page 11 lines 7-10)

(15) was the difference significant ? K-W test?

Yes, significant difference was observed (p<0.01). The detailed information concerning statistical analysis data was added in the revised manuscript as follows: "K-W test, p<0.01" (see page 11 line 22).

(16) the term "mechanistic relationship" is too vague. If methylation is active, is it expected that HgT$_f$ and MeHg$_f$ are correlated? (was it the case in your previous studies ? Meng et al 2014+ check literature). Then, if methylation is an important process at the artisanal mining site only, you can state it and try to explain why it is, and why not at the other site.

We believe that a restricted supply of newly deposited Hg to depths below the soil-water interface is a plausible explanation for the sharply reducing concentration of MeHg with depth at Gouxi; newly deposited Hg is constrained to surface soil and cannot be transferred to lower depth. Therefore, a direct positive relationship between HgT$_f$ and MeHg$_f$ concentrations in soil pore water was observed at Gouxi during the rice growing season (page 18 lines 24-28).

The Wukeng site has received significant historic Hg deposition as a function of large scale mining, but is not currently receiving significant inputs of fresh Hg. Atmosphere-derived mercury is physically unstable and bioavailable when it first enters the rice paddy (Hintelmann et al., 2002; Schuster, 2011). Immediate reactions of this new Hg with soil constituents are governed by adsorption-desorption interactions with soil surfaces (Schuster, 1991), which favour the retention of Hg in the surface layers of the soil profile. Over time this newly deposited Hg will be transformed into more stable, less available forms (Schuster, 1991), and the net methylation potential of this Hg will consequently decrease. The relatively low MeHg concentration in soil at Wukeng is indicative of old Hg which has become tightly bound to soil complexes over time, and is unavailable for methylation (Hintelmann et al., 2002). Consequently, there is no correlation between HgT$_f$ and MeHg$_f$ in soil pore water at Wukeng (see page 19 lines 1-10).

(17) I also believe this, but please insert references supporting this, as observed in other paddy field studies. Also, although it is implicitly stated, complete the sentence by reminding that after Day 80 the field is no more flooded, and hence methylation is probably stopped. -- edit OK I see now it is discussed a little bit further. Then, this sentence should be (re)moved, so that all interpretation is put together (not a bit here, and the rest further in the text)

**We definitely agree with the reviewer's comments here. We have deleted this sentence in the revised manuscript. More detailed explanation was added in the revised manuscript as follows:**

**Changing redox parameters over the rice growing season may affect the process of Hg methylation. Previous studies have observed that in artificially Hg-polluted soil, Hg bioavailability for methylation can be significantly affected by the level of water saturation (Rothenberg and Feng, 2012; Wang et al., 2014; Peng et al., 2012). Peng et al. (2012) specified that intermittent flooding, as opposed to continuous flooding, could reduce soluble Hg concentrations and inhibit Hg methylation in the rice rhizosphere, subsequently decreasing the accumulation of MeHg in rice grain. Flooded conditions enhance anaerobic microbial activities and increase MeHg yields. The drying of a paddy field is an important cultivation step to control rice plant tillering and increase yield. Therefore, one possible reason for the considerably elevated MeHg concentrations in soil at Gouxi between Day 20 and Day 80 relative to Day 100 is an enhancement of Hg bioavailability and numbers of SRB under flooded conditions that stimulated Hg methylation, and increased the soil MeHg concentration (Wang et al., 2014). As the paddy field dried from Day 80, some degree of net MeHg degradation may have occurred, which could be attributed to the decreased SRB numbers and proportion of Hg methylators in the rhizosphere under aerobic conditions (Wang et al., 2014). This could have contributed to a decreasing trend in soil MeHg concentration during the harvest period. (see page 14 lines 1-15).**

(18) be more specific about the processes. E.g. redox conditions change when the file dis dried, therefore MeHg degradation occurs. + give references of the biochemical processes taking place.

**Yes, we re-organized this sentence in the revised manuscript as follows: As the paddy field dried from Day 80, some degree of net MeHg degradation may have occurred, which could be attributed to the decreased SRB numbers and proportion of Hg methylators in the rhizosphere under aerobic conditions (Wang et al., 2014). This could have contributed to a decreasing trend in soil MeHg concentration during the harvest period. (see page 14 lines 12-15)**

(19) these references are relevant to a certain extent, but treated forest soils only. Are there references specific to paddy fields for this aspect ?

**Yes, the authors agree with the reviewer's comments here. We have reworked the model in the revised manuscript (see section 3.4 and response to comment #1). The references mentioned in this comment were**

**removed from manuscript.**

(20) not sure that "native" is the appropriate word. Native makes me think to geogenic i.e. "natural" Hg, while here it is mostly from anthropogenic sources. Consider rephrasing.

**Yes, the "native Hg" was change to "old Hg" in the revised manuscript. Furthermore, "old Hg" was defined as "Mercury present in the soil is termed 'old Hg', which can be either of geogenic and anthropogenic origin" in the revised manuscript (page 15 line 5-6).**

(21) is this water accumulated in the rice ??? then 34 cm seems a lot ! but unfortunately I cannot check Lan et 2010. The amount of transpiration seems very, very low if I compare e.g. to Brunel et al (1992) WRR 28 (5):1407-1416.

**We are very sorry for our unwary mistake. After re-checked this data from the cited literature, we found that $M_d$ is very low and needn't to be considered in this model. Hence, we removed $M_d$ from equation 4. Furthermore, we re-calculated the data in the revised manuscript (page 15 lines 24-26 and page 16 lines 1-6).**

(22) which? apart from runoff, what else could it be ? draining the paddy field ?

**$M_o$ is the cumulative amount of water lost by other pathways (e.g. animal activities and draining the paddy during the ripening period). We reworked this sentence in the revised manuscript. (see page 16 lines 3-4)**

(23) MAJOR COMMENT.
You need to take into account that the pool of 'old' Hg is probably constituted (partly) by deposition and irrigation from previous years. The Hg balance that you implemented compares 1 year of Hg input via deposition and irrigation to Hg accumulation over XX years. Is this the purpose of the Hg balance model? What kind of useful information does this bring?

**Yes, we definitely agree with the reviewer's wise comments here. After careful consideration, we re-worked the model in the revised manuscript (see detail in section 3.4 and the response to comment #1).**

(25) what if the Wukeng soil had had a low pH more favorable to methylation ? The important difference of pH between the two sites prevents in my opinion any conclusion regarding a possible difference between old / fresh Hg available for methylation. If the redox and pH conditions are not good for methylation, it will not occur (whatever the 'type' of Hg present in the soil). I recommend to discuss this, and reformulate the conclusions taking this into account.

**The reviewer has a good point here but we have interpreted this comment is the same was as comment #2. We trust that our response to point 2 is sufficient to respond to the reviewer's concerns here.**

(26) this Table contains a lot of information but is difficult to read. One cannot easily see to which sample matrix each result belongs (I think the text in left column should be vertically aligned to the top - this might already improve the readability but please try to improve this Table).

**Yes, we re-worked Table 2 based on the reviewer's comments in the revised manuscript (see page 28 lines 1-7 and page 29 lines 1-6)**

(27) increase slightly axis tick label and legend font size.

**Yes, we re-worked Figure 2 according the reviewer's comments in the revised manuscript (page 33 lines 1-4)**

[revised manuscript text omitted]

---

## Author Response (AR2)

**Response to comments from reviewers 1**

General comment:

The authors have done an excellent job responding to the critiques. The addition of microbial and biogeochemical explanations have greatly improved this manuscript.

**On behalf of my co-authors, I sincerelythank the anonymous reviewer for dedicating their time to provide further comments and criticism. My co-authors and I have made appropriatechanges to the manuscript. Revisionsare shown in red color in the revised manuscript, with our point-by-point response presented as follows.**

My general concern is that the model presented in Section 3.4 is still unnecessary. The authors' response to my comments and Reviewer #2 comments did not adequately justify its need and only present a simplified model. I argue it is not needed at all. The authors should consider removing the model or describe how Section 3.4 provides insight not provided by other sections.

**Yes, the authors agree with the reviewer's comment here. After carefully considering the comments herein in combination with those of Reviewer 2, the model presented in Section 3.4 (The relative mercury flux of different vectors to the soil Hg pool) was deleted in the revised manuscript.**

Lastly, I believe the authors should end Section 3.6 with a few concluding sentences about future work that may/should be done.

**Yes, we re-organized the Section 3.6 and a briefly concluding sentences concerning future work was shown in the revised manuscript as follows:**

**The relationship between MeHg and fresh deposited Hg implies that the concentration of Hg in ambient air could be used as a monitoring tool to assess the relative risk of MeHg production in the rice paddy environment, and the possible risk to human health that may be associated with the**

**accumulation of this MeHg in rice grain. However, we cannot distinguish between newly deposited Hg and old Hg stored in paddy soil over decades and ongoing research is necessary to continue to develop an improved understanding of Hg dynamics in rice paddy soils. When comparing relative risk between different vectors for Hg contamination (i.e. small-scale or historic large-scale mining), quantification of the pool of Hg available for methylation is critical to estimating relaible methylation rates. Ongoing work is urgently needed to further ascertain the relative importance of newly deposited Hg versus in situ Hg to the bioavailabile pool of Hg that can be methylated in rice paddy ecosystems. (see page18 lines 9-18)**

Technical corrections:

Page 2, Line 22: Change "a" to "an".

**Yes, "a" was changed to "an" in the revised manuscript. (page 2 line 22)**

Page 14, Line 12: Change "As the paddy field dried from Day 80," to "As the paddy field dried beginning on Day 80,". The phrase "from Day 80" is ambiguous.

**Yes, we re-organized this sentence as follows: "As the paddy field dried beginning on Day 80…." in the revised manuscript. (page 14 line 12)**

**Response to comments from reviewers 2**

The authors have made a substantial effort in the revision and the manuscript has greatly improved. I have only a few minor comments.

**The authors are most thankful the reviewer for dedicating time to review this manuscript and provide further comments and criticism, which will definitely improve this manuscript. We have completed a substantial revision and our changes are highlighted in red color in the revised manuscript. Our point-by-point response to the reviewer's comments is as follows:**

p.14, l.12: "from Day 80 on"

**Yes, we re-worked this sentence in the revised manuscript as follows: "As the paddy field dried beginning on Day 80….." (see page 14 line 12)**

p.15, l.13-17: Fp, Fw in the equations but Rp, Rw in the text.

p.16, l.5: cf. previous comment on the fraction of transpiration vs. evaporation in total ET. For example, Hossen et al. (2012) reported 64-70% of transpiration in total ET for two rice species in Bangladesh. It is very difficult for me to agree on the numbers Me = 40 cm and Mt = 1.4 cm (even though the influence on the model output is null because these values are added.) Please clarify, or update the values. Lan et al. 2010 is in Chinese, which I cannot read, so I cannot refer to their paper to understand more how this water balance was obtained.

**We agree with the reviewer's comments here. After carefully considering Reviewer 1's comments in combination with those of Reviewer 2, the model presented in Section 3.4 (The relative mercury flux of different vectors to the soil Hg pool) was deleted in the revised manuscript.**

p.18, l.6: rephrase for ex. "Active Hg methylation within the Gouxi rice paddy is implied in the present study, even though data directly supporting this hypothesis are lacking. Further work may help to bring more confidence on that particular point."

**Yes, we re-organized these sentences as follows: "Active Hg methylation within the Gouxi rice paddy is implied in the present study, even though data directly supporting this hypothesis are lacking. Further work may help to bring more confidence on that particular point" (page 16 line 1-3)**

Figue 4 caption "season on" (space)

**Yes, a space was added in the revised manuscript (page 32 line 2)**

Table 4 and various places in the text: I suggest to change everywhere "native Hg" to "old Hg" to be consistent throughout the manuscript.

**Yes, "native Hg" was changed to "old Hg" throughout the manuscript. (page 1 line 22, page 17 line 4, page 17 line 24)**

[revised manuscript text omitted]